# THE OPTIMIZATION LANDSCAPE OF SGD ACROSS THE FEATURE LEARNING STRENGTH

**Alexander Atanasov**[*,1,2,4], **Alexandru Meterez**[*,2], **James B Simon**[*,3,🐝], **Cengiz Pehlevan**[2,4,5]

[1] Department of Physics, Harvard University

[2] School of Engineering and Applied Science, Harvard University

[3] Department of Physics, University of California, Berkeley

[4] Center for Brain Science, Harvard University

[5] Kempner Institute, Harvard University

[🐝] Imbue

{atanasov,cpehlevan}@g.harvard.edu

## ABSTRACT

We consider neural networks (NNs) where the final layer is down-scaled by a fixed hyperparameter $\gamma$. Recent work has identified $\gamma$ as controlling the strength of feature learning. As $\gamma$ increases, network evolution changes from "lazy" kernel dynamics to "rich" feature-learning dynamics, with a host of associated benefits including improved performance on common tasks. In this work, we conduct a thorough empirical investigation of the effect of scaling $\gamma$ across a variety of models and datasets in the online training setting. We first examine the interaction of $\gamma$ with the learning rate $\eta$, identifying several scaling regimes in the $\gamma$-$\eta$ plane which we explain theoretically using a simple model. We find that the optimal learning rate $\eta^*$ scales non-trivially with $\gamma$. In particular, $\eta^* \propto \gamma^2$ when $\gamma \ll 1$ and $\eta^* \propto \gamma^{2/L}$ when $\gamma \gg 1$ for a feed-forward network of depth $L$. Using this optimal learning rate scaling, we proceed with an empirical study of the under-explored "ultra-rich" $\gamma \gg 1$ regime. We find that networks in this regime display characteristic loss curves, starting with a long plateau followed by a drop-off, sometimes followed by one or more additional staircase steps. We find networks of different large $\gamma$ values optimize along similar trajectories up to a reparameterization of time. We further find that optimal online performance is often found at large $\gamma$ and could be missed if this hyperparameter is not tuned. Our findings indicate that analytical study of the large-$\gamma$ limit may yield useful insights into the dynamics of representation learning in performant models.

## 1 INTRODUCTION

The study of large-width limits of neural networks (NNs) has led to a variety of insights about deep learning. As network width $N$ tends to infinity, networks in either "PyTorch standard" parameterization or neural tangent kernel (NTK) parameterization converge to kernel methods (Jacot et al., 2018; Lee et al., 2019; Sohl-Dickstein et al., 2020). Kernel methods have a fixed set of features, and networks in this kernel limit have fixed representations at each layer over training. By contrast, realistic neural networks adapt their features during the course of training (Fort et al., 2020; Vyas et al., 2022; Lee et al., 2020). Feature learning is widely believed to be intimately connected to the excellent performance of NNs on practical tasks.

If one takes a centered NN[1] $f(\boldsymbol{x}; \boldsymbol{\theta})$ in NTK parameterization with inputs $\boldsymbol{x}$ and parameters $\boldsymbol{\theta}$ and scales the output as $\tilde{f}(\boldsymbol{x}; \boldsymbol{\theta}) \equiv \alpha f(\boldsymbol{x}; \boldsymbol{\theta})$, one can control the degree to which the NN learns features. Taking $\alpha \to \infty$ yields the "lazy limit," where the network behaves as a kernel (Chizat et al., 2019). By contrast, if one sets $\alpha = (\gamma\sqrt{N})^{-1}$ for fixed $\gamma$ and simultaneously scales the learning rate as $\eta N$, one obtains a network whose hidden representations evolve even at infinite width (Geiger et al., 2020; Mei et al., 2018; Rotskoff & Vanden-Eijnden, 2018; Yang & Hu, 2021; Bordelon & Pehle-

---

[*] Joint first authorship, alphabetized

[1] In this paper, we assume all networks are centered, meaning that $f(\boldsymbol{x}; \boldsymbol{\theta}) = 0$ at initialization. This can be achieved by subtracting a non-trainable copy of the network at initialization.

van, 2022), a scaling termed the "maximal update parameterization" or "$\mu$P" by Yang & Hu (2021). Although at practical widths and dataset scales, NNs in standard and NTK parameterization deviate from their infinite-width kernel limit (Lee et al., 2020), networks in $\mu$P achieve their infinite width behavior at realistic scales (Vyas et al., 2023a; Noci et al., 2024). Networks parameterized in $\mu$P generally achieve better performance, and $\mu$P can be utilized effectively for zero-shot hyperparameter transfer from small models to large (Yang et al., 2021; Bordelon et al., 2023). The width does not affect the degree of feature learning in $\mu$P, so we identify $\gamma$ as the *feature learning strength* or *richness parameter* (Woodworth et al., 2020).

This story leaves $\gamma$ as a free hyperparameter to be tuned, but despite its importance, little is practically known about the effects of varying $\gamma$. In this paper, we perform a thorough scaling analysis in which we train a variety of deep networks with $\gamma$ ranging across many orders of magnitude, with an eye towards both optimal tuning of $\gamma$ and cataloging the diverse dynamical phenomena that occur in different hyperparameter regimes. In addition to studying the lazy ($\gamma \ll 1$) and rich ($\gamma \sim 1$) regimes, we pay particular attention to what we term the *ultra-rich* ($\gamma \gg 1$) regime.

In this work, we focus on the *online* setting, where a fresh batch of data is given at each step and we train for a fixed time set by a compute budget. This is realistic of modern large-scale language and vision models and is scientifically clarifying, removing the confounding effect of finite dataset size.

Concretely our contributions are as follows. We systematically vary $\gamma$ over a suite of models spanning MLPs, CNNs, ResNets, and Vision Transformers (ViTs) on a variety of datasets. In all settings:

- Sweeping over both $\gamma$ and the learning rate $\eta$, we observe a characteristic phase portrait of model performance in the $\gamma$-$\eta$ plane, which depends only on the choice of the loss function and not the model architecture (Figure 1). We show that the optimal learning rate scales as $\eta_* \sim \gamma^2$ in the lazy ($\gamma \ll 1$) regime and $\eta_* \sim \gamma^{2/L}$, where $L$ is the model depth, in the ultra-rich ($\gamma \gg 1$) regime. Our phase portrait reveals an upper limit on $\gamma$ for a deep network to be optimizable for a given compute budget.

- After adopting the correct learning rate scaling, we find that large $\gamma$ networks usually exceed or match the performance of naive $\gamma = 1$ networks if trained for sufficiently long (Figure 2). This disagrees with findings in the offline training setting (Petrini et al., 2022), where strong feature learning was found to degrade performance.

- At small $\gamma$, we observe the scaling of $\eta$ with $\gamma$ predicted by kernel theory. To our knowledge, we are the first to observe that transformer-based architectures on large datasets can empirically reach the lazy limit. At slightly-too-large $\eta$ in the lazy ($\gamma \ll 1$) regime, we observe the "catapult effect" of Lewkowycz et al. (2020) in which the loss quickly grows large before gradually converging. We see catapults for $\gamma^2 \lesssim \eta \lesssim \gamma$ for cross-entropy loss, but only in a narrow band around $\eta \sim \gamma^2$ for MSE. Smaller $\gamma$ leads to larger catapults.

- At large $\gamma$, we observe *silent alignment* as in Atanasov et al. (2022): at early times, the kernel aligns itself to the task before the loss drops. During this early period, NNs exhibit strikingly similar dynamics upon rescaling $\tau = \eta t/\gamma$, with the loss often exhibiting stepwise drops as in (Saxe et al., 2014; Simon et al., 2023b) even in very realistic networks. We further observe that the loss drops at the end of silent alignment coincide with the Hessian growth predicted by the edge of stability (EOS; Cohen et al. (2021)).

- At optimal learning rate, we see a surprising agreement in the loss trajectories of large-$\gamma$ networks at *late times* in training. This suggests that large-$\gamma$ networks are learning similar features and functions past a certain threshold. We verify this similarity by comparing losses, accuracies, learned functions, and internal representations.

- Finally, we reproduce and analytically derive our phase portrait and many of the above behaviors in the simple setting of a linear neural network model (Section 4). There, we show the modified $\eta$ scaling in the large $\gamma$ regime is due to a progressive sharpening effect.

We review the catapult, silent alignment, and progressive sharpening effects in Appendix C.

## 1.1 RELATED WORKS

Chizat et al. (2019) introduced the laziness parameter $\alpha$ in their seminal work. Mei et al. (2018); Rotskoff & Vanden-Eijnden (2018) introduced networks in mean-field parameterization. Geiger

et al. (2020) introduced the correct scaling of $\alpha$ with $N$ as $\alpha = (\gamma\sqrt{N})^{-1}$ to achieve feature learning at infinite width. They also highlighted that $\gamma$ can be identified as the parameter that controls feature learning. Yang & Hu (2021) expanded this idea to deep networks of arbitrary architecture and coined the term $\mu$P. Bordelon & Pehlevan (2022) described this limit in terms of dynamical mean field theory. Extensions to infinite depth were performed in Bordelon et al. (2023) and Yang & Littwin (2023) and to infinite attention in Bordelon et al. (2024c). We give a review of network parameterizations in Appendix B. A key motivation for this work is that $\gamma$ is the main free parameter in the DMFT description of infinite-width feature learning networks, warranting a more thorough empirical investigation.

The question of optimal $\gamma$ was explored in the paper of Petrini et al. (2022). There, for several models in the offline setting, they showed large $\gamma$ networks performed worse. In a linear network toy model, Pesme et al. (2021) showed that SGD noise could be absorbed into a rescaling of $\alpha$. Sclocchi et al. (2023) performed a large set of thorough experiments to study the interplay between the SGD noise in different regimes of $\gamma$ and find that SGD noise can hurt or help in the offline setting. By contrast to these papers, we focus on the *online setting*. We comment on this further in Section 2.1. Under cross-entropy loss, Agarwala et al. (2020) performed extensive sweeps on un-centered networks over $\gamma, \eta$, identifying $\gamma$ as the soft-max temperature, similarly finding that slightly larger $\gamma$ achieves top performance. Our cross-entropy results overlap with this work and recover their findings, but we additionally identify a different way to scale $\eta$ at large $\gamma$, allowing us to probe the very rich limits without training instability.

Various analytical studies of the large-$\gamma$ limit have been performed in the setting of linear networks (Atanasov et al., 2022; Tu et al., 2024), with Jacot et al. (2021) terming this limit the "saddle-to-saddle" regime. The feature learning parameter has been shown to play a key role in the mechanism for grokking in (Kumar et al., 2024). The paper of (Kalra et al., 2023) performed a thorough analysis of the causes of progressive sharpening and catapult behavior in networks across $\mu$P and SP in the offline setting. In the online setting we show that all effects are present in $\mu$P and are modulated by the $\gamma$ parameter. In the setting of a linear network, (Marion & Chizat, 2024) also found a nontrivial and depth-dependent sharpening effect. Earlier works studying the effect of varying the $\gamma$ parameter (Sclocchi et al., 2023; Bordelon & Pehlevan, 2022; 2023) focused on the *offline* setting of training with repeated data until some convergence criterion is satisfied.

## 2 SETUP AND NOTATION

We train neural networks $f(\boldsymbol{x}; \boldsymbol{\theta})$ parameterized using $\mu$P scaling, with the output further down-scaled by a factor of $\gamma$.[2] We denote the width of a given network by $N$ and the depth of a feed-forward network by $L$.[3] We train online on batches $\mathcal{B}_t = \{\boldsymbol{x}_\mu, \boldsymbol{y}_\mu\}_{\mu=1}^B$ batch with size $B$ for $t \in \{1, \dots, T\}$ time steps. The loss of each batch is given by a mean of per-example losses:

$$L_{\mathcal{B}_t}(\theta) = \frac{1}{B} \sum_{\mu \in \mathcal{B}_t} \ell\left(\frac{1}{\gamma} f(\boldsymbol{x}_\mu; \boldsymbol{\theta}), \boldsymbol{y}_\mu\right). \tag{1}$$

We will consider both mean-squared-error and cross-entropy loss for $\ell$. In this work we focus on optimizing the parameters by vanilla SGD, as even in this simple setting, the effect of $\gamma$ is not well-understood. For small batch sizes, the the SGD effects are controlled by the ratio $\eta/B$ (Jastrzebski et al., 2017; Smith et al., 2020; Sclocchi et al., 2023). We verify this this in Appendix K.

We study networks trained on the datasets, MNIST, CIFAR and TinyImageNet. Our motivation is to study networks training in the online setting over several orders of magnitude in time. To this end, we adopt larger versions of these datasets: "MNIST-1M" and CIFAR-5M, and apply strong data augmentation to TinyImagenet. We generate MNIST-1M using the denoising diffusion model (Ho et al., 2020) in Pearce (2022). We use CIFAR-5M from Nakkiran et al. (2021). Earlier results in Refinetti et al. (2023) show that networks trained on CIFAR-5M have very similar trajectories to those trained on CIFAR-10 without repetition.

### 2.1 WHY TRAIN ONLINE?

In the modern era of large models, the full training set is seldom repeated (Kaplan et al., 2020; Muennighoff et al., 2023). Online, the effect of SGD noise was seen to have negligible effect on

---

[2] We verify our $\mu$P implementation, confirming consistency of dynamics across width in Appendix I.

[3] Here, depth counts the number of weight matrices, e.g. $f(\mathbf{x}) = W_2\sigma(W_1\mathbf{x})$ has depth $L = 2$.

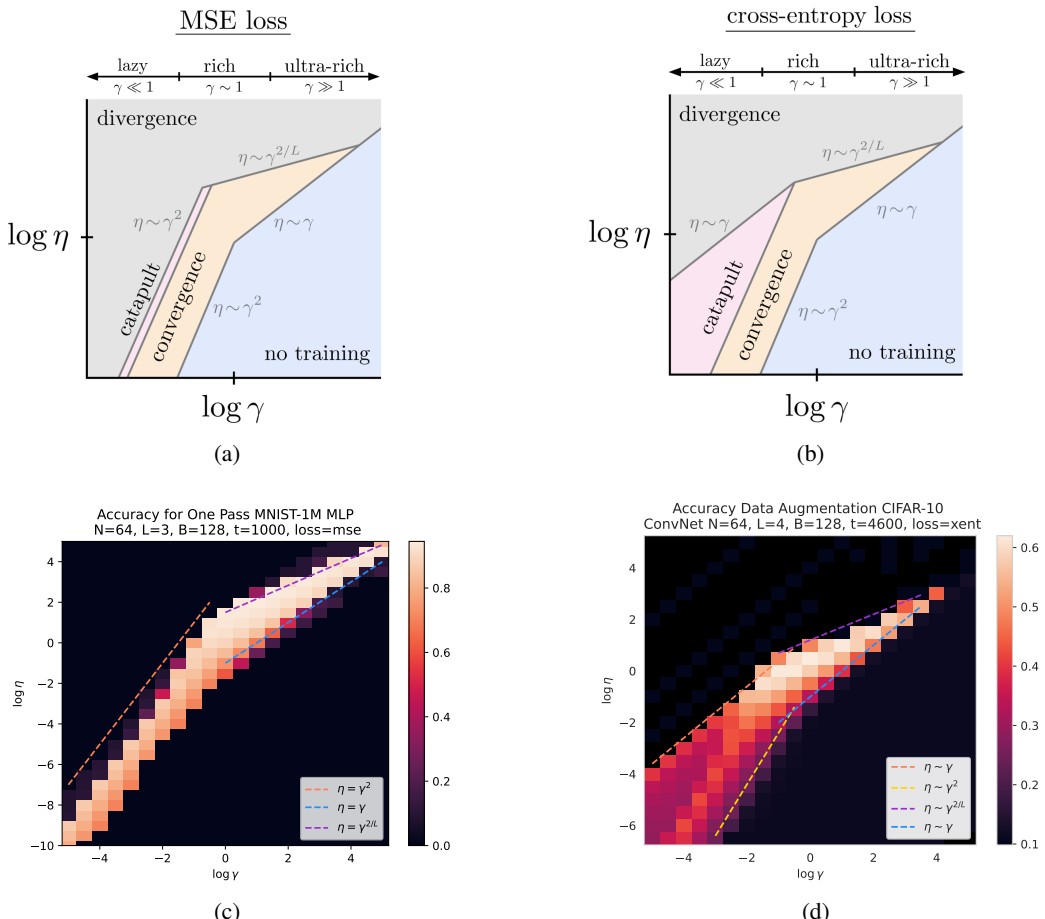

Figure 1: **Phase portraits of the $\gamma - \eta$ plane for MSE and cross-entropy losses.** a,b) Schematics of the regimes of network training for both losses. As the number of gradient steps increases, the lower boundaries of the convergent region descend, and the "no training" region shrinks. $L$ here denotes the network depth. All scalings are obtained analytically in Section 4. c,d) Final accuracies of deep networks trained across a grid of values of $\gamma, \eta$ for c) and MLP on MNIST-1M with MSE and d) a CNN on CIFAR-5M with cross-entropy loss. As shown by dashed lines, these empirics agree with our analytical diagrams. These results are robust to the choice of model and task.

scaling both empirically (Vyas et al., 2023b) and theoretically (Paquette et al., 2024). The online setting thus allows us to isolate properties of $\gamma$ as an optimization hyperparameter for the population gradient flow that SGD approximates. This removes the complicated overfitting effects that have been theoretically shown to build up when data is repeated (Bordelon et al., 2024a), as well as the many regimes of SGD under an interpolation constraint (Sclocchi & Wyart, 2024).

## 3  EMPIRICAL RESULTS

In this section we consider a variety of architectures: MLPs, CNNs, ResNet-18s, and Vision Transformers (ViTs), trained on a variety of tasks. We first discuss the different regimes observed in the $\eta$-$\gamma$ plane across architectures. We then discuss the properties of the Hessian, and identify specific dynamical phenomena. Finally, we focus on the large $\gamma$ limit and show that learned functions learned representations largely agree across large $\gamma$ networks.

### 3.1  PHASE PORTRAIT OF $\eta$ WITH $\gamma$

We begin by considering the $\gamma$-$\eta$ plane for a variety of architectures and datasets. We sweep jointly over every pair of $\eta, \gamma$ in a log-spaced grid running from $\gamma = 10^{-5}$ to $10^5$. For each $\gamma$, we sweep from $\eta = 10^{12}$ to $\eta = 10^{-12}$ *downwards* in a log-spaced grid until the first convergent $\eta$ is reached.

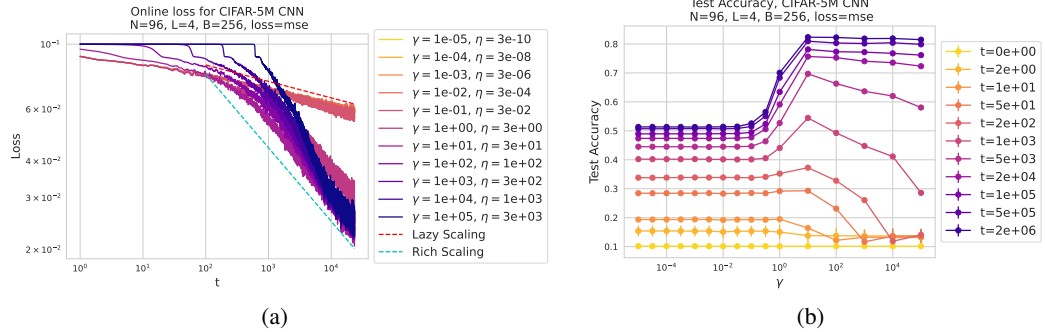

Figure 2: a) Online loss curves of CNN trained on CIFAR-5M across $\gamma$, at the second largest convergent $\eta$. $\eta$ scales as $\gamma^2$ for $\gamma < 1$ and $\gamma^{2/L}$ for $\gamma > 1$. We also observe that very large $\gamma$ have a long period of flat loss before a drop. We overlay dashed lines to highlight the different power law scalings of loss with training time observed in the lazy and rich regime. b) Final test accuracy as a function of $\gamma$. We see that, as long as one trains long enough, larger $\gamma$ yields equal or better generalization to $\gamma = 1$, but the returns are marginal past some point. We verify this for other networks in Appendix L.3. Error bars show the small variance over three initializations.

After finding the first convergent $\eta$, we sweep over a range of $\eta$ below it. We consider both MSE and cross-entropy losses. Further details are given in Appendix A.

**At small $\gamma$, $\eta \propto \gamma^2$ for lazy networks**   We consider the limit of small $\gamma$, $\gamma \to 0$. This recovers the lazy limit of Chizat et al. (2019). The (empirical) neural tangent kernel of the network is $K_{NTK} \equiv \nabla f \cdot \nabla f \sim \Theta(\gamma^{-2})$. In kernel regression under MSE loss, the maximum achievable learning rate is proportional to the top eigenvalue of the Hessian. This sets the upper bound to be $\eta \sim \gamma^2$. More generally at small $\gamma$ the Hessian is dominated by the Gauss-Newton term, as we will explicitly write in Section 3.2, equation 3. This term scales as $\Theta(\gamma^{-2})$. Moreover, in order to achieve a trainable network as $\gamma \to 0$, one needs the loss to have a nonzero change in $\mathcal{L}$ in that limit. We have:

$$\mathcal{L}_{t+1} - \mathcal{L}_t = -\eta |\nabla_\theta \mathcal{L}|^2 = -\eta \left| \frac{1}{B} \sum_{\mu=1}^B \frac{1}{\gamma} \nabla_\theta f \, \ell' \left( \frac{1}{\gamma} f(\boldsymbol{x}_\mu, \boldsymbol{\theta}), \boldsymbol{y}_\mu \right) \right|^2. \qquad (2)$$

By centering, near initialization $\ell'$ is $\Theta_\gamma(1)$. The left hand side is thus $\Theta(\gamma^{-2})$, so the minimum learning rate to see the loss drop in $T$ steps scales as $\eta \sim \gamma^2/T$. We see this on the left region of the plots in Figure 1(c) and 1(d). We verify that networks of different but sufficiently small $\gamma$ have the same loss dynamics in Appendix L.1, indicating that the lazy limit is reached by realistic networks.

**At small $\gamma$ and larger $\eta$, catapults can occur**   For $\eta$ larger than the sharpness allowed by convex optimization, the loss explodes early on in time. For a range of $\eta$ above this, the loss eventually drops again. This is the catapult effect of Lewkowycz et al. (2020). The maximal learning rate for these scales as $\gamma^2$ under MSE loss and as $\gamma$ under cross-entropy loss. We revisit this in Section 3.3 and Figure 4(a).

**At large $\gamma$, there are two learning rate scalings**   Empirically, in Figures 1(c) and 1(d), we observe a "triangle of optimizability" in the large $\gamma$ limit. To our knowledge prior works have not highlighted this fact. The upper boundary of the maximal learning rate before the loss explodes is found empirically to scale as $\eta \sim \gamma^{2/L}$. On the other hand, at large $\gamma$, activation movement precedes function movement, by contrast to the lazy limit. The minimal learning rate scaling in order for the activations to move by $\Theta_\gamma(1)$ is $\eta \sim \gamma$, as in (Bordelon & Pehlevan, 2022), or $\eta \sim \gamma/T$ under a budget of $T$ training steps. Taking $C \propto TB$ to be the compute budget, we see that generally increasing $C$ increases the optimizable region for large $\gamma$.

**Sufficiently large $\gamma$ NNs see improved scaling laws**   We plot the loss in time across $\gamma$ in Figure 2(a). We overlay two different scaling curves as dashed lines. We see the slopes between the

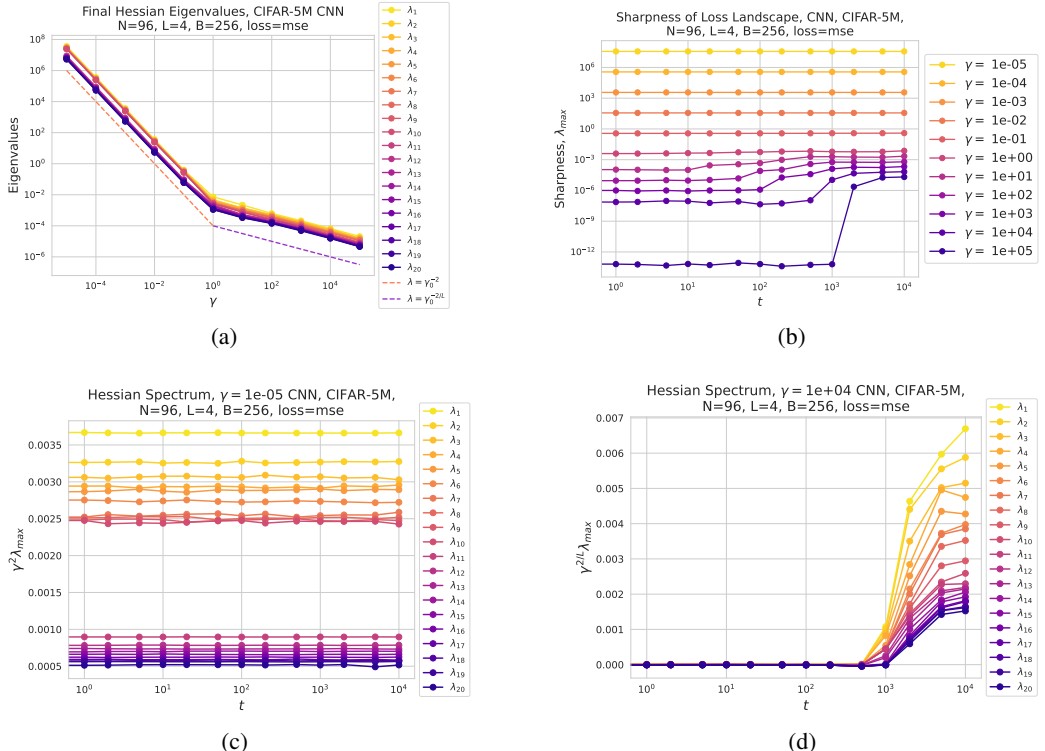

(a)                 (b)

(c)                 (d)

Figure 3: a) The top 20 eigenvalues of the Hessian at the end of training vs $\gamma$ for an NN trained with MSE. We clearly see two regimes. For small $\gamma$ we see that $\lambda_{\max}$ b) The eigenvalue vs time across several value of $\gamma$. c) The top eigenvalues across time for a lazy network. In the lazy setting, we see ten outliers, equal to the number of classes. The Hessian otherwise does not change in MSE. d) The same for a rich network. At late time, rather than seeing a small set of outliers, we see many eigenvalues grow to a sizeable range. Further plots are given in Appendix M.

small $\gamma$ lazy networks and the large $\gamma$ rich networks appear substantively different. The former slope can be predicted from kernel theory, as in Pillaud-Vivien et al. (2018) and Bordelon & Pehle-van (2021). Increasing $\gamma$ beyond some threshold does not appear to improve the scaling law. This echoes similar findings as found by the recent model of Bordelon et al. (2024b).

**Large-$\gamma$ NNs achieve good generalization, given sufficient training time** We plot the accuracy of NNs across $\gamma$ in Figure 2(b). We see that generalization improves with larger $\gamma$, eventually reaching a roughly constant value. This contrasts with studies in the offline setting (Geiger et al., 2020; Sclocchi et al., 2023) where performance decreases at large $\gamma$. We highlight that if networks do not train long enough, a non-monotonicity is indeed present, but this goes away with longer online training.

### 3.2 HESSIAN SCALING

We next consider the scaling of the Hessian of the loss as a function of $\gamma$. On a batch $\mathcal{B}$ of $B$ test points, this is given by $\mathcal{H}_{\mathcal{B}} = \nabla_\theta^2 L_{\mathcal{B}}$, or:

$$\mathcal{H}_{\mathcal{B}} = \underbrace{\frac{1}{B}\sum_{\mu=1}^{B} \nabla f(\boldsymbol{x}_\mu) \cdot \nabla f(\boldsymbol{x}_\mu)\, \ell''(f(\boldsymbol{x}_\mu), y_\mu)}_{\mathcal{G}} + \underbrace{\frac{1}{B}\sum_{\mu=1}^{B} \nabla^2 f(\boldsymbol{x}_\mu)\, \ell'(f(\boldsymbol{x}_\mu), y_\mu)}_{\mathcal{R}}. \tag{3}$$

Here, we have split the Hessian into a Gauss-Newton term $\mathcal{G}$ and a loss gradient term $\mathcal{R}$. The Hessian contains important properties about the optimization landscape. One particularly important

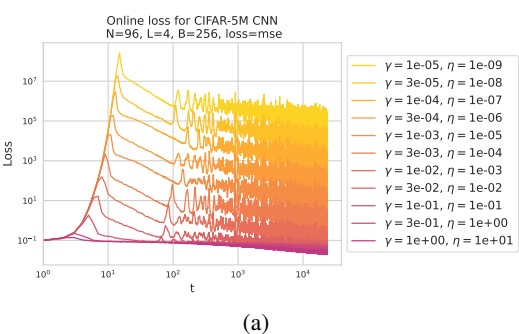 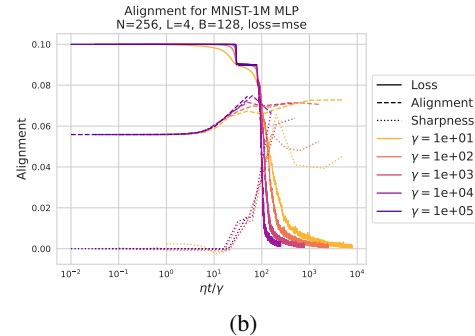

(a)                                    (b)

Figure 4:   a) At sufficiently small $\gamma$, we observe that the loss catapults. Small $\gamma$ incur a larger catapult. Further catapult plots are in Appendix N. b) At large $\gamma$, we do not observe catapults. At the optimal learning rate, the loss stays near a saddle for an extended period of time, before suddenly dropping. We show that during this period, the alignment between the final-layer kernel and the task grows before the loss drops. Kernel-target alignment is defined in Appendix P. Further silent alignment plots are in Appendix Q. We also see characteristic step-wise loss drops in this regime.

quantity is top eigenvalue $\lambda_{max}$, also known as the **sharpness**. The sharpness governs the maximal allowable learning rate in a convex optimization problem.

Across all tasks we find remarkably consistent behavior of the Hessian spectrum, see Figure 3(a). For $\gamma \ll 1$, the network stays lazy and the Hessian does not evolve. There, the Hessian is dominated by the $\mathcal{G}$ term, which scales as $\gamma^{-2}$. Once $\gamma$ exceeds a threshold however, we observe a distinct change in scaling at the end of training. For feed-forward MLPs and CNNs, we verify a scaling going as $\gamma^{-2/L}$. In section 4, we derive this scaling in a linear network model.

We also track how the top eigenvalues of the evolve in time in Figures 3(b) to 3(d). For lazy networks we predictably see no evolution. In the next section we show that small $\gamma$ but larger $\eta$ networks can catapult and lower their sharpness. For rich networks we highlight that a very extensive number of eigenvalues grow to large scale. This is in contrast with the commonly observed phenomenon that for a $C$-class classification task there should be $C$ outliers. Here, in the lazy limit there are indeed 10 distinct outliers, but in the rich regime there are no distinct outliers at late times.

### 3.3  DYNAMICAL PHENOMENA

We here describe a catalog of different dynamical phenomena that we find in networks trained in the small- and large-$\gamma$ regimes. We give background on these effects in Appendix C.

**The catapult effect at small $\gamma$**   We see in Figure 4(a) that small $\gamma$ networks exhibit large increases followed by decreases in the loss, with remarkably regular behavior, decreasing in scale as $\gamma$ grows. In Appendix D, we leverage a similar linear network model to arrive at the desired catapult scalings with $\gamma$ under both MSE and cross-entropy loss.

**Silent alignment at large $\gamma$**   We now consider the dynamics at large $\gamma$. Empirically, we find that our sweeps do not detect any catapult effects once $\gamma$ is large enough. Taking the optimal learning rate for SGD at large $\gamma$ still yields a long loss plateau. During this plateau, the NN can be shown to be adapting its features, and appears to align its hidden-layer representations with the task. We illustrate this in Figure 4(b). This effect was identified in linear networks at small initialization in Atanasov et al. (2022), and shown to empirically hold in a restricted class of NNs on whitened data. Here, we show that it arises in a much broader class of realistic settings.

**Stepwise loss drops at large $\gamma$**   At very large $\gamma$, we often observe loss trajectories following a characteristic staircase pattern, with one or more plateaus punctuated by sudden drops. This is a prediction of linear network theory (Saxe et al., 2014; Simon et al., 2023a) that seems to hold even for very realistic models. We see these step-wise trajectories for MLPS in Figures 4(b), 31 and 41,

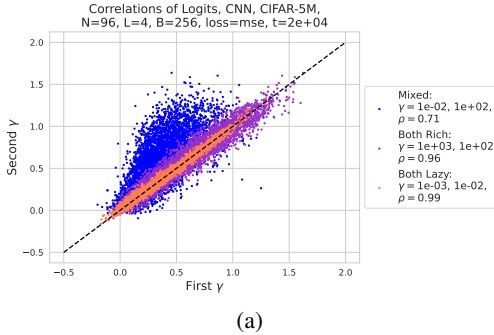 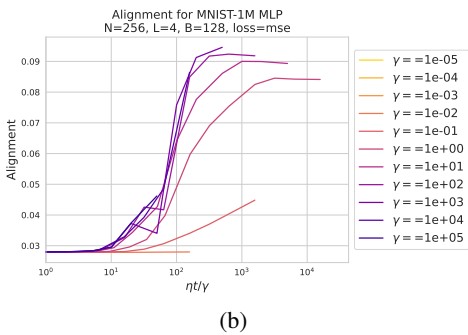

(a)             (b)

Figure 5: a) Function outputs between pairs of networks, as described in the main text. We see that lazy networks have nearly identical function outputs, and that rich networks agree in their function outputs at the end of training as well. See Appendix R for further plots. b) We study the kernel-target alignment of final-layer representations across $\gamma$ at the end of training.

CNNs in Figures 31 and 40 and ViTs in Figure 16. We show that, upon rescaling time as $\tau = \eta t/\gamma$, the early time dynamics coincide. The first loss drop goes as $t \sim \gamma$. We explain the origin of this timescale in Section 4.

**Progressive sharpening at the end of silent alignment**    Across all tasks and models, we find that the loss drop that ends silent alignment is always accompanied by a growth in the sharpness to the final value of approximately $2/\eta$. We show this in Figure 4(b). This is the large $\gamma$ analogue of the progressive sharpening effect observed in Cohen et al. (2021). This effect is not present at small $\gamma$.

### 3.4   Comparing Learned Functions and Representations

Finally, we consider the question of to what extend different networks are learning the "same function" and the same representations. To this end, we consider two distinct networks A, and B as well as a held out set of test points. For each point $\boldsymbol{x}_\mu$, there is a correct class $c$, so that $[\boldsymbol{y}_\mu]_{c'} = \delta_{c,c'}$ as a one-hot vector. We consider the value that a given network places on $[\boldsymbol{y}_\mu]_c$. On one axis, we plot network and on the other we plot network B. Networks are learning the same function if and only if this plot yields an exact diagonal line. In the lazy limit, we see in Figure 5(a) that indeed the two networks are learning nearly identical functions. We also see strong agreement between different network in the rich limit at the end of training. By contrast, rich and lazy networks do not have substantial function agreement.

We then consider the learned representations in Figure 5(b). Here, we plot a version of centered kernel alignment (CKA) between the final layer network activation kernel $K$ and the "task kernel" $\boldsymbol{y}\boldsymbol{y}^\top$ for $\boldsymbol{y}$ a $P \times C$ class label matrix. We see that, after adopting a suitable time rescaling, the alignment scores for a variety of networks across $\gamma$ agree. Larger $\gamma$ networks retain slightly higher alignment scores.

## 4   A Simple Model Explaining Observed Scalings

In this section, we provide a theoretical model that explains all observed scaling relationships empirically observed in Figure 1 and analytically reproduces our phase portraits.

The following simple model will prove sufficient: a deep linear network of width one, trained on a single example. The network function in this case is $f(x) = w_L \ldots w_1 x$. We will initialize all weights to 1 at initialization, and due to the commutativity of multiplication they will receive the same gradients and remain equal throughout training. We are thus justified in replacing all weights with a single weight $w$ and, letting $x = 1$, we need only keep track of the function value $f = w^L$.[4]

---

[4]A technicality: because the variable $w$ is repeated $L$ times, it will receive a gradient $L$ times larger than one of the original variables $w_\ell$. To recover the original training dynamics, we would thus need to downscale $\eta$

| MSE Loss | | | | | Cross-Entropy Loss | | | | |
|---|---|---|---|---|---|---|---|---|---|
| Lazy ($\gamma \ll 1$) | | | Ultra-rich ($\gamma \gg 1$) | | Lazy ($\gamma \ll 1$) | | | Ultra-rich ($\gamma \gg 1$) | |
| $\eta_{\min}$ | $\eta_{\text{crit}}$ | $\eta_{\max}$ | $\eta_{\min}$ | $\eta_{\max}$ | $\eta_{\min}$ | $\eta_{\text{crit}}$ | $\eta_{\max}$ | $\eta_{\min}$ | $\eta_{\max}$ |
| $\frac{\gamma^2}{T}$ | $\gamma^2$ | $\gamma^2$ | $\frac{\gamma}{T}$ | $\gamma^{2/L}$ | $\frac{\gamma^2}{T}$ | $\gamma^2$ | $\gamma$ | $\frac{\gamma}{T}$ | $\gamma^{2/L}$ |

Table 1: SGD learning rate scalings for our toy models for two loss functions in the lazy and ultra-rich regimes. All values come out of our one-parameter model except $\eta_{\text{crit}}$ for MSE loss, which comes from the two-parameter model given in Appendix D. Predicted scalings match NN experiments.

We divide the function value by $\gamma$ and center it by subtracting off its value at initialization before passing it to the loss function.

## 4.1 SCALINGS FOR MSE LOSS

We first consider MSE loss with a target value $y = 1$. All together, this yields a loss equal to

$$\mathcal{L}_{\text{MSE}}(w) = \frac{1}{2}\left(\frac{1}{\gamma}(w^L - 1) - 1\right)^2. \tag{4}$$

with $w = 1$ at initialization. The loss is minimized by

$$w_* = (\gamma + 1)^{\frac{1}{L}} \approx \begin{cases} 1 + \frac{\gamma}{L} & \text{for } \gamma \ll 1, \\ \gamma^{\frac{1}{L}} & \text{for } \gamma \gg 1, \end{cases} \tag{5}$$

where we use "$\approx$" to indicate that we are neglecting higher-order terms in $\gamma$ as appropriate to the regime. In the lazy $\gamma \ll 1$ regime, $w_*$ is close to the initial $w$, whereas in the ultra-rich $\gamma \gg 1$ regime, the parameter $w$ must grow substantially. The Hessian of the loss at convergence is then:

$$\mathcal{L}''_{\text{MSE}}(w_*) = \frac{L^2}{\gamma^2}w_*^{2L-2} \approx \begin{cases} \frac{L^2}{\gamma^2} & \text{for } \gamma \ll 1, \\ L^2\gamma^{-2/L} & \text{for } \gamma \gg 1. \end{cases} \tag{6}$$

**Maximal convergent learning rates:** In order to converge, the learning rate must satisfy $\eta < 2/\mathcal{L}''(w^*)$ near convergence. Thus, from Equation (6), we see that the maximal convergent learning rate $\eta_{\max}$ will scale as

$$\eta_{\max} \sim \begin{cases} \gamma^2 & \text{for } \gamma \ll 1, \\ \gamma^{2/L} & \text{for } \gamma \gg 1. \end{cases} \tag{7}$$

**Minimal nontrivial learning rates:** In investigating the minimal learning rates, we can safely take the gradient flow approximation. We define the error $\Delta = -\mathcal{L}'(w) = 1 - \frac{1}{\gamma}(w^L - 1)$.

When $\gamma \ll 1$, we may linearize the network output with respect to $w$, and doing so we find that $w \approx 1 + \frac{\gamma}{L}(1 - \exp(-\frac{L^2\eta T}{\gamma^2}))$. The error is then given by $\Delta \approx \exp(-\frac{L^2\eta T}{\gamma^2})$. Therefore, in order for the loss to move a nontrivial amount, we require the scaling $\eta_{\min} \sim \gamma^2/T$.

When $\gamma \gg 1$, in order for the network to learn features, we must have that the hidden layer activations move by $\Theta_\gamma(1)$. Because in this model these are given by $w^\ell x$ for each layer $\ell$, we see we require $\Theta_\gamma(1)$ weight movement. The weight updates are given by $\dot{w} = -\eta\mathcal{L}'(w) = \frac{\eta L}{\gamma}w^{L-1}\Delta$. So in order for the network to learn features in $T$ steps of training, we require $\eta > \eta_{FL} \sim \gamma/T$.

Beyond constraining that the activations move, we also consider how long it takes to escape the initial saddle and lower the loss. Using that $w'(t) \approx \frac{\eta L}{\gamma}w^{L-1}$ during the initial period, we find that

$$w(T) \approx \begin{cases} e^{\frac{L\eta T}{\gamma}} & \text{if } L = 2, \\ \left(1 - \frac{L\eta T}{\gamma}\right)^{\frac{-1}{L-2}} & \text{if } L \geq 3. \end{cases} \tag{8}$$

---

by a factor of $L$. However, because we assume $L = \Theta(1)$, we neglect this for simplicity and our $\gamma, T$ scalings will be unaffected.

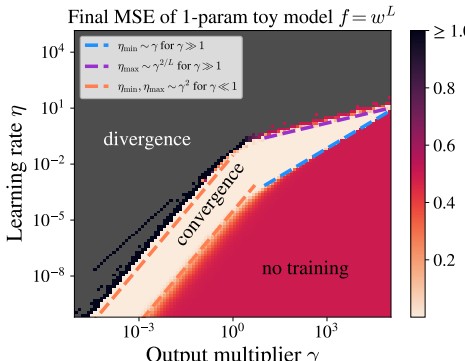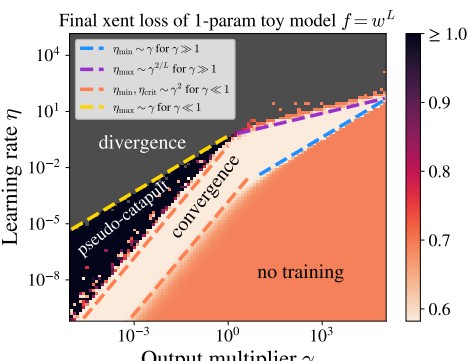

Figure 6: Simulations of our one-parameter model with MSE loss (left; see Equation (4)) and cross-entropy loss (right; see Equation (9)) with varying $\gamma, \eta$ recover our principal phase portrait. Simulations run with $L = 5$. For simulations of a two-parameter model which exhibits true catapults, see Figures 8 and 9.

This phase ends when the function value $\frac{w^L}{\gamma}$ reaches order unity. In order for this to occur, we need $\eta \sim T^{-1}\gamma \log \gamma$ if $L = 2$ and $\eta \sim \gamma/T$ if $L \geq 3$. Neglecting the logarithmic factor, we find a lower bound $\eta_{\min} \sim \frac{\gamma}{T}$. Note as a consequence that the early-time dynamics are approximately $\gamma$-invariant after the rescaling $\tau \equiv \eta t/\gamma$. We verify this in Figure 4(b), 5(b), and Appendix L.2.

**Critical learning rate for the catapult effect:** The above one-parameter model is sufficient to obtain all the coarse dynamical phenomena we see except for so-called loss "catapults." As described by (Lewkowycz et al., 2020), a network trained in the lazy regime with a just-too-large learning rate will grow quickly in loss, but then move to a flatter region of parameter space and converge.

The catapult effect generally requires two parameter dimensions to describe (Damian et al., 2023), so our one-parameter model cannot capture it. In Appendix N, we describe and simulate a minimal *two*-parameter model that exhibits catapults. We find that for MSE loss, catapults occur in a narrow band $\eta_{\text{crit}} < \eta < \eta_{\max}$, with $\eta_{\text{crit}} \sim \gamma^2$ just like $\eta_{\max}$.

## 4.2 SCALINGS FOR CROSS-ENTROPY LOSS

We now consider the toy cross-entropy loss

$$\mathcal{L}_{\text{xent}}(w) = \frac{\log(e^{\tilde{f}(w)} + 1)}{1 + e^{-1}} + \frac{\log(e^{-\tilde{f}(w)} + 1)}{1 + e}, \tag{9}$$

where we use the shorthand $\tilde{f}(w) = \gamma^{-1}(w^L - 1)$ for the centered and rescaled function output. This is a binary cross-entropy loss with ground-truth class probabilities $p_0 = (1 + e)^{-1}$ and $p_1 = 1 - p_0$ chosen so that the loss minimum occurs at $\tilde{f}(w) = 1$. The resulting loss landscape is, scaling-wise, the same as that for MSE, with one salient difference: when $\tilde{f}(w) \gg 1$, we have $\mathcal{L}_{\text{xent}}(w) \sim |\tilde{f}(w)|$ as opposed to $\mathcal{L}_{\text{MSE}} \sim \tilde{f}^2(w)$.

The main effect of this difference is seen in the lazy ($\gamma \ll 1$) regime when $\eta$ is too large for stability at the minimizer. Once $w - 1$ has grown to order just larger than $\gamma$ (and we are still seeing linearized dynamics), it begins to experience *constant-sized* restoring steps $\delta w \sim -\frac{\eta L}{\gamma} \text{sign}(w - 1)$.[5] It is only if $\eta \sim \gamma$ or larger that this step is large enough to lead to a $\Theta_\gamma(1)$ change in $w$, upon which we escape our initial linear region and can diverge. The result is that the maximum *stable* learning rate for our toy model in the lazy regime is $\eta_{\text{crit}} \sim \gamma^2$, but the maximum *nondivergent* learning rate is $\eta_{\max} \sim \gamma$. In our minimal two-parameter model described in Appendix D, we find that the loss does eventually settle back down and converge for $\eta_{\text{crit}} < \eta < \eta_{\max}$, and that this triangle of the phase diagram is there a true catapult regime. All the above scalings are summarized in Table 1.

---

[5]Contrast this with the *linear-sized* restoring steps $\delta w \sim -\frac{L^2}{\gamma^2}\delta w$ which one sees with MSE.

We perform simulations of our one-parameter model in a fashion analogous to our parameter sweeps for deep networks for $T = 10^3$ steps of gradient descent. The results, plotted in Figure 6, confirm our analytical scalings and match our phase portraits for deep networks (e.g. Figure 1). Finally, we also provide a similar analysis in the case of the Adam optimizer in Appendix Appendix F.

## 5 CONCLUSION

We have presented a set of large-scale empirical sweeps across $\gamma$ for deep networks trained on realistic data. We have empirically identified how the upper bound for optimizable $\eta$ is determined across a host of different $\gamma$ in terms of the training time and the depth of the network. We see that, conditional on an adjusted learning rate scaling with $\gamma$, larger $\gamma$ NNs are competitive with or better than $\gamma = 1$ NNs after training long enough. We have seen very predictable scaling of hessian spectra with $\gamma$ and identified various dynamical phenomena whose appearance is governed by the size of $\gamma$. By appeal to a linear NN, we have been able to explain many of our observed scalings. In future steps, we hope to to extend our analysis to optimizers beyond SGD.

### AUTHOR CONTRIBUTIONS

AA ran early empirics that revealed the phase plot structure, initiating the project, ran all MNIST-1M and most CIFAR-5M experiments, wrote the initial draft, offered feedback on the solvable model, and led the team logistically. AM ran all ViT experiments and experiments on TinyImageNet, helped with CIFAR-5M experiments, observed the role of attention temperature in the final phase diagram, and offered technical feedback on both experimental and theoretical fronts. JS proposed and developed the solvable model exhibiting all phases, ran empirics on that model, and offered technical feedback on all experiments. CP supervised the project and gave feedback and direction. All authors contributed substantially to the final draft.

### CODE AVAILABILITY

The code to reproduce all figures in this paper can be accessed at:

https://github.com/Pehlevan-Group/Richness_Sweep.

### ACKNOWLEDGEMENTS

The authors are thankful to Blake Bordelon, Daniel Kunin, Katherine Lee, Allan Raventos, Jascha Sohl-Dickstein, Jacob Zavatone-Veth, and Brian Zhang for helpful discussions. AA thanks Blake Bordelon and Jeremy Cohen for in-depth discussions that motivated many of the questions investigated in this paper. We thank Blake Bordelon and Jacob Zavatone-Veth for comments on a draft. AA was supported by a Yaser Abu-Mostafa Fellowship from the Fannie and John Hertz Foundation. AM acknowledges the support of a Kempner Institute Graduate Research Fellowship. JS gratefully acknowledges support from the National Science Foundation Graduate Fellow Research Program (NSF-GRFP) under grant DGE 1752814. C.P. is supported by NSF grant DMS-2134157, NSF CAREER Award IIS-2239780, DARPA grant DIAL-FP-038, a Sloan Research Fellowship, and The William F. Milton Fund from Harvard University. This work has been made possible in part by a gift from the Chan Zuckerberg Initiative Foundation to establish the Kempner Institute for the Study of Natural and Artificial Intelligence. Discussions contributing to this work took place at the Kavli Institute for Theoretical Physics' 2023 program on "Deep Learning from the Perspective of Physics and Neuroscience." AA and JS thank the builders of a particular house in Reston, VA which, to their surprise, they discovered they had both lived in at different times.

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

## A    EXPERIMENTAL DETAILS

We evaluate all our experiments on A100 and H100 GPUs, using PyTorch. In order to compute the Hessian eigenvalues we use PyHessian (Yao et al., 2020) evaluated on 1 fixed batch of size $512$ at every point in time. In the following subsections we provide individual experimental details for every model trained.

Across all models, we swept over $\gamma$ from $10^{-5}$ to $10^{5}$ in a log-spacing with two samples per decade. We swept over $\eta$ from $10^{10}$ down to a lower limit that varied by experiment, as reported on the plots. For each run that caused a gradient explosion, the model was not saved. We only saved models with $\eta$ less than or equal to the first convergent learning rate for each value of $\gamma$. For the MNIST-1M plots and CIFAR-5M plots, at each value of $\gamma$ we only kept the seven values of $\eta$ that are within a factor of $10^{3}$ from the top stable $\eta$ to make the plots as in e.g. Figure 1.

In all cases, we created a CenteredModel class, ensuring that the output of the network is zero at initialization by making the following definition for the trained function $\tilde{f}$:

$$\tilde{f}(\boldsymbol{x}, \boldsymbol{\theta}) \equiv \frac{1}{\gamma} \left[ f(\boldsymbol{x}, \boldsymbol{\theta}) - f(\boldsymbol{x}, \boldsymbol{\theta}_0) \right]. \tag{10}$$

Here, $\boldsymbol{\theta}_0$ is the initial setting of the parameters, and is not trainable.

In the case of ViTs we compare training with QK-LayerNorm and QK-Norm in order to account for the attention logit temperature.

### A.1    DETAILS ON THE PARAMETERIZATION

On top of the centering operation explained above, ee parameterize all our networks under $\mu$P (and Depth-$\mu$P, respectively) parameterization. More concretely, we define our $L$ layer neural network of width $N$ $f(x)$ with nonlinearity $\phi(x)$ as:

$$h^{1}(x) = \frac{1}{\sqrt{D}} W^{0} x \tag{11}$$

$$h^{\ell+1}(x) = \tau h^{\ell}(x) + \frac{1}{\sqrt{N} L^{\alpha}} W^{\ell} \phi(h^{\ell}(x)) \tag{12}$$

$$f(x) = \frac{1}{\gamma N} W^{L} \phi(h^{L}(x)) \tag{13}$$

where $D$ is the input dimension, and for $\tau = 0$ and $\alpha = 0$ we recover $\mu$P, and for $\tau = 1$ and $\alpha = \frac{1}{2}$ we recover Depth-$\mu$P (Bordelon et al., 2023; Yang & Littwin, 2023). When running SGD, we update the weights with a global learning rate going as $\eta N$. That is:

$$W_{t+1}^{\ell} = W_{t}^{\ell} - N \eta \nabla \mathcal{L} \tag{14}$$

Similar modifications exist for Adam, see (Yang & Littwin, 2023).

In the case of Transformers, we additionally change the attention temperature to $\frac{1}{d_q}$ following (Yang et al., 2022). We further detail and contrast this parameterization with the NTK parameterization in Appendix B.

### A.2    COMMENTS ON NUMERICAL STABILITY IN THE LAZY LIMIT

Here we detail a ubiquitous problem that arose in the training of lazy neural networks at small $\gamma$. To our knowledge, this problem has not been highlighted in the literature. At very small values of $\gamma$, approximately $10^{-3}$, the appropriate learning rate in the lazy limit scales as $\eta \sim \gamma^2$. This leads to weight updates on the order of $10^{-6}$ or less. Machine epsilon for float32 is near $10^{-7}$. Because the final output of the function involves the subtraction in equation 10, the weight updates matter in both relative and absolute magnitude. For this reason, we find that we needed to go to double precision to reliably obtain curves below $\gamma = 10^{-3}$. In practice, we find that a value of $\gamma = 10^{-3}$ or even $10^{-2}$ is essentially enough to reach the lazy limit in the online setting for essentially all tasks listed. See Appendix L.1 for a check of this.

We also highlight, that it is insufficient to simply zero out the last layer weights at initialization as a way to center the network. The lazy limit of a network with zero-initialized last layer weights yields the final-layer conjugate kernel or NNGP kernel. This kernel has very different properties from the NTK, especially in that its dimensionality scales as the width rather than the number of parameter of the network.

## A.3 MNIST-1M

MNIST-1M example digits

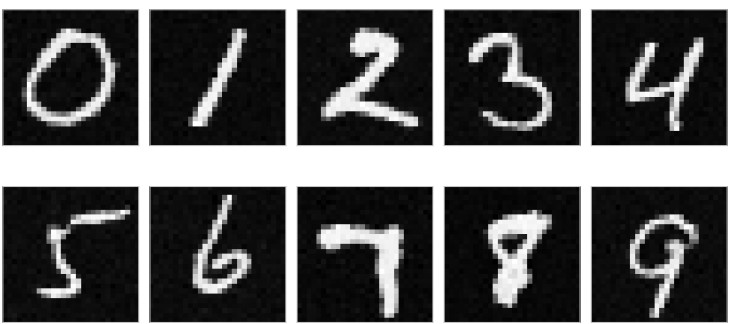

Figure 7: Examples of digits from the MNIST-1M dataset, with one digit from each class generated.

A diffusion model generated version of MNIST with 1 million images, which we will call MNIST-1M, was used to train these networks. We swept over a variety of widths, depths, batch sizes, and architectures. Each individual run took approximately 15 seconds on an A100 GPU. For each setting of width, depth, batch size, and architecture, we swept over the fine grid of $\gamma, \eta$ stated above. This led to 7 runs per value of $\gamma$ or approximately 150 runs total for each phase plot. Most runs were repeated over three initialization seeds, and key quantities, such as loss and accuracy, were seen to minimally vary over the initialization. This is expected in $\mu$P, by arguments of approximate initialization independence at sufficiently large widths as in (Bordelon & Pehlevan, 2023; Mei et al., 2019).

The architectures we used were MLPs of depths $2, 3, 4$ and a feed-forward CNN of depth $4$ consisting of two convolutional and two feed-forward layers.

The corresponding test set for this dataset is the original MNIST test set.

Although it is quite easy for most networks to get to near 100 percent accuracy in one pass through this dataset, we believe that it fills an important gap in the space of datasets. It provides a task that is fast enough to learn, allowing for quick iteration, yet avoiding repetitions of data so as to remain online. In the era of online training, it is useful to have simple but large-scale datasets in order to test theory against. We do not claim that this dataset is representative of realistic imagenet scale natural data, but it is encouraging that virtually all effects observed in the CIFAR-5M and tinyimagenet datasets are also present in this smaller setting.

We observe that training accuracy on MNIST-1M meaningfully transfers to the original MNIST, and there is no observed overfitting effect in the networks that we've trained.

## A.4 CIFAR-10 AND CIFAR-5M

We train on the full dataset of CIFAR-5M introduced in (Nakkiran et al., 2021). This includes both the train set of 5 million images as well as the 1 million image test set, making for a total of 6 million images. We stay in the one pass setting. We also trained on CIFAR-10 with heavy data augmentation. We swept over width, depth, batch size, and architecture as above. In essentially all plots, we followed the same recipe for sweeping over $\gamma$ and $\eta$. For some of the ResNets and Vision transformers, larger sweeps were performed.

The architectures we used were feed-forward CNNs of depth $4$ as above, ResNets of varying depths as well as ResNet-18s, and Vision Transformers.

The corresponding test set for this dataset is the original CIFAR-10 test set.

## A.5    TINYIMAGENET

In order to train on TinyImagenet in an online setting, we use random data augmentations: rotations and crops. All the experiments on this dataset have been run using batch size $128$. Individual architectural details are offered for every plot. We train all the models under $\mu P$ (or Depth-$\mu P$ where specified), using stochastic gradient descent (SGD).

## B    REVIEW OF NTK PARAMETERIZATION AND $\mu$-PARAMETERIZATION

For detailed discussion of $\mu$P more generally, see (Yang & Hu, 2021; Yang et al., 2021; Geiger et al., 2020; Bordelon & Pehlevan, 2022). The aim of this section is conceptual clarity on the difference between NTK parameterization and $\mu$-parameterization, as well as to give some degree of motivation of the latter. See also the recent accessible works of (Yang et al., 2023) and (Chizat & Netrapalli, 2023).

There are several equivalent ways of parameterizing NNs that will yield the same dynamics as $\mu$P. In what follows, we focus on a scalar-output feed-forward network for simplicity. The key distinction between NTK and $\mu$ parameterizations it that the hidden layer activations move $\Theta_N(1/\sqrt{N})$ in the former and $\Theta_N(1)$. One can straightforwardly extend the argument to more general architectures.

### B.1    NTK

For simplicity, we take all layers to have hidden width $N$ and the input space to have dimension $D$. We Given an example $\boldsymbol{x}_\mu$, the pre-activation $\boldsymbol{h}_\mu^{\ell+1}$ in layer $\ell+1$ is given by in terms of the activation $\phi^\ell$ in layer $\ell$ as

$$\boldsymbol{h}_\mu^{\ell+1} = \frac{1}{\sqrt{N}} \boldsymbol{W}^\ell \cdot \boldsymbol{\phi}_\mu^\ell, \tag{15}$$

where $\phi^\ell$ is given by an element-wise nonlinearity an element-wise non-linearity $\sigma$ acting on $\boldsymbol{h}_\mu^\ell$. The $N^{-1/2}$ factor enforces that $\boldsymbol{h}_\mu^{\ell+1}$ to be $\Theta(1)$ in root-mean-square norm at initialization as $N \to \infty$. The output of the network $f_\mu$ is given by

$$f_\mu = \frac{\alpha}{\sqrt{N}} \boldsymbol{w}^L \cdot \boldsymbol{\phi}_\mu^L. \tag{16}$$

In NTK parameterization $\alpha$ is taken to be order $1$. It is the laziness parameter identified in (Chizat et al., 2019). The change in the function is given by

$$\frac{df_\mu}{dt} = -\eta_{SP} \sum_\nu K_{\mu\nu} \ell'(f_\nu, y_\nu). \tag{17}$$

Here $K_{\mu\nu} = \nabla_\theta f_\mu \cdot \nabla_\theta f_\nu$ is the NTK between data points $\mu$ and $\nu$. The NTK is easily seen to be order $\alpha^2$ and $\ell'$ is order $1$ at small $\alpha$. In order to have the change in the function be $\Theta(1)$ we set $\eta_{SP} = \eta_0/\alpha^2$.

By applying the chain rule, one obtains that the preactivations move as (Geiger et al., 2020; Jacot et al., 2018; Chizat & Bach, 2018)

$$\frac{d\boldsymbol{h}^\ell}{dt} \sim \eta_{SP} \frac{\alpha}{\sqrt{N}} = \frac{\eta_0}{\alpha\sqrt{N}}. \tag{18}$$

Thus, at large $N$ and $\alpha = 1$ the pre-activations of this network evolve as $\Theta(N^{-1/2})$. As $N \to \infty$ this becomes a kernel limit, given by the infinite-width NTK that does not learn features.

## B.2   $\mu$P

In order to get $\Theta_N(1)$ preactivation and activation movement, we see that we must take $\alpha = 1/\gamma\sqrt{N}$ for a constant $\gamma$, fixing $\eta_0$ to be $N$-independent. This implies that we simply replace the final layer of the network by:

$$f_\mu = \frac{1}{\gamma N} \boldsymbol{w}^L \boldsymbol{\phi}_\mu^\ell, \quad \eta_{SP} = N\eta. \tag{19}$$

As the prior analysis shows, this still yields that $df_\mu/dt \sim Theta_N(1)$ at initialization. Note that we have not considered how to scale $\eta$ with $\gamma$. The $\gamma$ scaling is the subject of the empirical studies of this paper.

## C   REVIEW OF DYNAMICAL PHENOMENA

In this section we review the essential dynamical phenomena studied in prior literature that arise in various regions of the $\eta$-$\gamma$ plane studied

### C.1   CATAPULT EFFECT

The original catapult effect was studied in (Lewkowycz et al., 2020). There, in the NTK parameterization, it was observed that at large but finite width, taking the learning rate $\eta$ slightly above the bound given by convex optimization theory $2/\lambda_{max}$ led to a regime where the sharpness of the loss monotonically decreased. Follow up work has observed that the sharpness itself can undergo catapult behavior (Kalra & Barkeshli, 2023; Kalra et al., 2023). In (Bordelon & Pehlevan, 2023) it was argued and shown that in $\mu$P, one can observe catapults even at infinite width.

### C.2   SILENT ALIGNMENT

In (Atanasov et al., 2022), it was argued that linear neural networks trained on whitened data starting from small initialization undergo an alignment of their weights in the task-relevant direction before their function output scale has grown sufficiently. The kernel alignment thus grows well before a loss drop, indicating that even at a loss plateau, the NN has learned a meaningful representation of the data. The results were empirically tested on a relatively small class of more realistic tasks. Here, we extend the class of tasks, and observe silent alignment phenomena for nonlinear networks on realistic and anisotropic tasks.

### C.3   EDGE OF STABILITY

In (Cohen et al., 2021), in the setting of full-batch gradient descent, it was observed that the maximum hessian eigenvalue grew to the scale of $2/\eta$, where it stabilized. This effect was further studied in (Damian et al., 2023). In the SGD setting, a similar effect persists, though the final sharpness is modified from the $2/\eta$ limit to something that depends more holistically on the spectrum. See (Agarwala et al., 2022) for a theoretical study and discussion in this setting.

## D   A TWO-PARAMETER MODEL RECOVERING CATAPULT SCALINGS

In this appendix, we describe a two-parameter model which recovers our main phase portrait and — unlike the one-parameter model of Section 4 — exhibits true catapult behavior with just-too-large learning rates in the lazy regime. We will first give an analytical treatment, then conclude with simulations.

Our model and argument will be very similar to those used by Lewkowycz et al. (2020), but with a slightly different focus: rather than examining the effect of the width $N$ on catapult behavior, we study how the $\gamma$ *parameter* controls catapult behavior. When width and $\gamma$ are properly disentangled (that is, when one uses $\mu$P), the width of the network plays no role in determining catapults. Indeed, recent work (Bordelon & Pehlevan, 2023) has shown that deep networks can catapult even at infinite width. The catapult effect does not happen at large $\gamma$, so we will ultimately focus on small $\gamma$.

We begin with a multi-parameter model which we will shortly simplify. We consider a two-layer linear network $f(\boldsymbol{x}) = \boldsymbol{W}_2 \boldsymbol{W}_1 \boldsymbol{x}$ with width $N$ trained on a single example $\boldsymbol{x}, \mathbf{y}$. We assume that $\|\boldsymbol{x}\|_2 = 1$, and then without loss of generality, we may assume the input is scalar with $x = 1$ due to the rotational symmetry of SGD. For MSE loss, we will assume the target has $\|\mathbf{y}\| = 1$, and we may likewise treat the output as scalar without loss of generality. For cross-entropy loss, we will simply assume the output is scalar.

Relabeling $\boldsymbol{u} = \boldsymbol{W}_2^T \in \mathbb{R}^N$ and $\boldsymbol{v} = \boldsymbol{W}_1 \in \mathbb{R}^N$, we now have a function value $f = \boldsymbol{u}^T \boldsymbol{v}$. We now make two observations. First, when $N$ is large, $\boldsymbol{u}$ and $\boldsymbol{v}$ will be orthogonal at initialization. Second, as noted by Lewkowycz et al. (2020), upon training by SGD, $\boldsymbol{u}$ and $\boldsymbol{v}$ will each remain in the two-dimensional subspace spanned by the two vectors at initialization. We are thus free to rotate into this subspace and assume $\boldsymbol{u}, \boldsymbol{v} \in \mathbb{R}^2$.

We will assume that, at initialization, $\boldsymbol{u} = [1, 0]$ and $\boldsymbol{v} = [0, 1]$. Now observe that, for any loss function $\mathcal{L}(f)$ depending only on $f$, it will be the case that $\nabla_{\boldsymbol{u}} \mathcal{L} \propto \boldsymbol{v}$ and $\nabla_{\boldsymbol{v}} \mathcal{L} \propto \boldsymbol{u}$. As a consequence, it will be the case that, at all later times, will will continue to have $\boldsymbol{v} = \texttt{swap}(\boldsymbol{u})$, where $\texttt{swap}$ is an operation that swaps the two elements of its argument. Denoting $\boldsymbol{u}(t) = [u_1(t), u_2(t)]$, we thus have $\boldsymbol{v}(t) = [u_2(t), u_1(t)]$. We are down to just two parameters, $u_1(t)$ and $u_2(t)$, with $u_1(0) = 1$ and $u_2(0) = 0$.

We denote the downscaled network output as $\tilde{f} = \frac{1}{\gamma} f$. Because $\tilde{f}|_{t=0} = 0$, we do not need to center it.

## D.1 ANALYTICAL ARGUMENT FOR CATAPULTS FOR MSE LOSS

In this section, we leverage an argument similar to Lewkowycz et al. (2020) to demonstrate how the $\gamma$ parameter controls catapult behavior for MSE loss. Let the loss be $\mathcal{L}_{\text{MSE}} = \frac{1}{2}(\tilde{f} - 1)^2$. We then study the discrete-time dynamics of gradient descent with learning rate $\eta$. Upon gradient updates, we have that

$$\begin{bmatrix} u_{1;t+1} \\ u_{2;t+1} \end{bmatrix} = \begin{bmatrix} u_{1;t} \\ u_{2;t} \end{bmatrix} - \frac{1 - \tilde{f}}{\gamma} \begin{bmatrix} u_{2;t} \\ u_{1;t} \end{bmatrix}. \tag{20}$$

We then define

$$K \equiv \frac{1}{\gamma^2} \left( \|\boldsymbol{u}\|^2 + \|\boldsymbol{v}\|^2 \right) = \frac{2}{\gamma^2}(u_1^2 + u_2^2); \qquad \Delta \equiv 1 - \tilde{f}, \tag{21}$$

where $K$ is the neural tangent kernel of the model and $\Delta$ is the residual. Using these substitutions, we find that the discrete-time dynamics yield

$$\begin{aligned} \tilde{f}_{t+1} - \tilde{f}_t &= \eta \Delta_t K_t + \frac{\eta^2}{\gamma} \Delta_t^2 \tilde{f}_t, \\ K_{t+1} - K_t &= \frac{\eta \Delta_t^2}{\gamma^2} \left[ \eta K_t + \frac{4 \tilde{f}_t}{\Delta} \right]. \end{aligned} \tag{22}$$

Suppose now that we choose $\eta$ too large such that $\tilde{f}$ begins to explode. When $\tilde{f}$ is large, we have that $\tilde{f}/\Delta \approx -1$. Observe now that, so long as $\eta K_t \le 4$, we find that $K$ will *decrease* upon subsequent gradient steps, eventually stabilizing the dynamics of $\tilde{f}$. We thus see that the upper bound on $\eta$ in the catapult regime scales as $\eta \sim \gamma^2$.

## D.2 CROSS-ENTROPY LOSS

As with the one-parameter model of Section 4, we may also train our two-parameter model with the binary cross-entropy loss given in Equation (9). The dynamics for cross-entropy loss are substantially more difficult to analyze than those for MSE — in fact, as noted by Meng et al. (2024), even an ordinary linear model trained with cross-entropy loss will exhibit *chaotic dynamic* in certain parameter regimes! In actual fact, the dynamics are quite similar, scaling-wise, to those for MSE, except in a critical "catapult triangle" in the lazy regime in which $\eta_{\text{crit}} < \eta < \eta_{\text{max}}$, where $\eta_{\text{crit}} \sim \gamma^2$ and $\eta_{\text{max}} \sim \gamma$. Recall that our one-parameter model neither converged nor diverged in this regime. Our two-parameter model begins to show similar behavior, but, after sufficiently long training time, bounces to a different part of the loss landscape and converges. The dynamics here are similar to

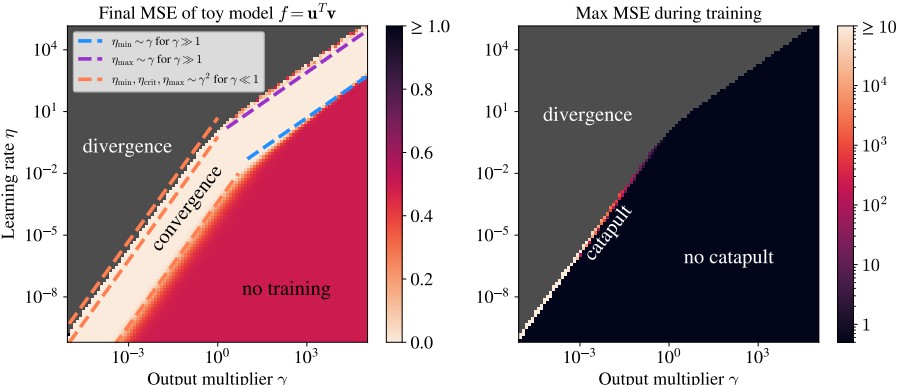

Figure 8: **Simulations of our two-parameter model with MSE loss recover all our scalings, plus catapult dynamics.** The left plot shows final loss, while the right plot shows the maximum loss reached during training. Grey pixels indicate divergent runs. Catapults occur along a narrow band. Note that, for this model, $L = 2$, so $\eta_{\min} \sim \gamma$ in the ultra-rich regime.

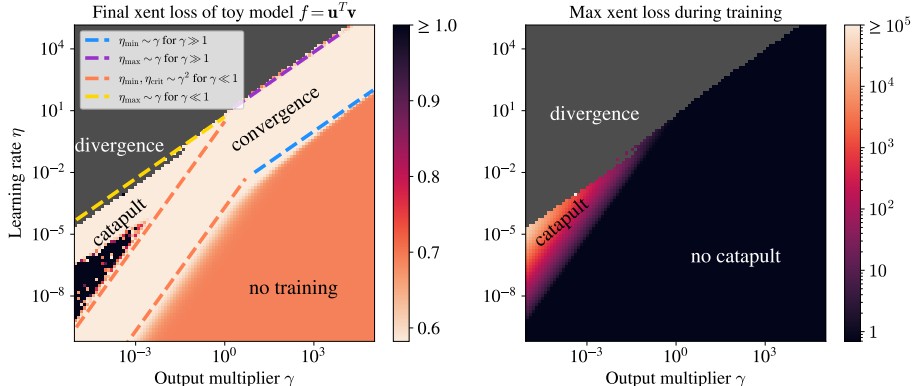

Figure 9: **Simulations of our two-parameter model with cross-entropy loss recover all our scalings, plus catapult dynamics.** The left plot shows final loss, while the right plot shows the maximum loss reached during training. Grey pixels indicate divergent runs. Catapult dynamics occur within a triangular region. The black subtriangle within the catapult triangle for final loss is made of runs which would eventually converge given substantially more training iterations. Note that, for this model, $L = 2$, so $\eta_{\min} \sim \gamma$ in the ultra-rich regime.

the catapult dynamics described by Damian et al. (2023), except that the "unstable direction" sees a straight-walled loss basin instead of a quadratic basin (think $\mathcal{L}(x) = |x|$ instead of $\mathcal{L}(x) = x^2$). We anticipate an analytical treatment in the style of Damian et al. (2023) would be elucidating, but it could make up a substantial part of a paper in and of itself, so at this point we turn to numerics.

### D.3    SIMULATIONS OF THE TWO-PARAMETER TOY MODEL

We numerically simulate our two-parameter model for both loss functions considered. For MSE, we train for $T = 10^3$ steps. For cross-entropy, we train for $T \approx 3 \times 10^4$ steps to give the dynamics in the "catapult triangle" more time to converge.

The results are shown in Figures 8 and 9. We plot both the final loss and the maximum loss at any point in training. In both cases, we see phase portraits very similar to those for our one-parameter model (Figure 6), but we also see the loss exploding before convergence in the catapult phase.

# E    SCALING RELATIONSHIPS ARE UNAFFECTED BY LEARNING RATE WARMUP AND DECAY

In this section, we empirically verify that adding a warmup and cosine decay to the learning rate, as common in more modern deep learning practice, does not affect the key scaling relationships observed. The models are trained using learning rate warmup for approximately 5% of the total number of steps, followed by a cosine decay until the end of training. All models are still trained with SGD. In the next section, we analyze the effect of alternative optimizers such as Adam and SignSGD.

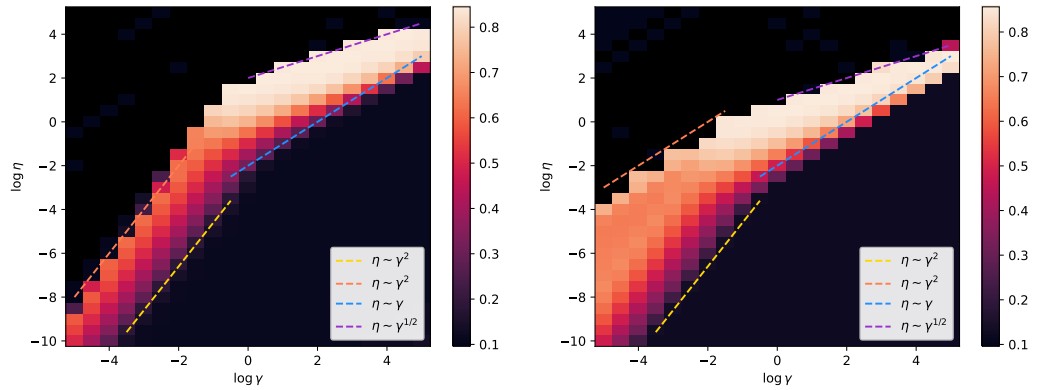

Figure 10: Empirical runs of a 3 layer and readout ConvNet without residuals model trained with SGD with MSE loss (left; see Equation (4)) and cross-entropy loss (right; see Equation (9)) with varying $\gamma, \eta$ on CIFAR-5m.

# F    SCALING RELATIONSHIPS FOR ADAM

All analysis thus far treats MSE loss. However, in practice, Adam is a more common choice of optimizer when training large models, so it is worth asking how our story changes when switching SGD to Adam. We repeat our analysis our toy model for SignSGD — which is equivalent to Adam without momentum — in Appendix F. We find a very similar story with different scaling exponents: $\eta_{\min}, \eta_{\max} \sim \gamma$ when $\gamma \ll 1$ and $\eta_{\min}, \eta_{\max} \sim \gamma^{1/L}$ when $\gamma \gg 1$. Remarkably, unlike with SGD, the upper and lower scaling exponents are the same in the ultra-rich regime ($\gamma \gg 1$) due to the fact that Adam escapes the saddle near initialization in $O(1)$ time. This suggests that Adam may be the optimizer of choice for future investigations into the ultra-rich regime.

In this appendix, we obtain both empirical and analytic phase plots and scaling relationships between $\eta$ and $\gamma$ under the Adam optimizer. We present a variant of the one-parameter model in Section 4 for training with SignSGD (Kingma, 2014). Adam can be thought of as essentially SignSGD with momentum. The predictions of the one-parameter model are then validated in via simulation.

## F.1    THEORETICAL PHASE PLOT FOR SIGNSGD

Adam is essentially SignSGD (Bernstein et al., 2018) with momentum, and the effect of this momentum will only change our story up to order-unity prefactors, so we instead study SignSGD. The update rule for SignSGD is

$$\theta \mapsto \theta - \eta \cdot \text{sign}\left(\nabla_\theta \mathcal{L}\right), \tag{23}$$

where $\theta$ is a vector of parameters and $\mathcal{L}$ is the loss. As in Section 4, we wish to determine the viable range of learning rates for training the centered, rescaled target function $\tilde{f}(w) = \gamma^{-1}(w^L - 1)$ on MSE and cross-entropy losses.

In the lazy regime ($\gamma \ll 1$), we wish the parameter $w$ to move by an amount scaling like [desired update] $\sim \gamma$. This parameter feels a gradient scaling as [gradient] $\sim \gamma^{-1}$, so with SGD, we required a learning rate scaling as $\eta \sim$ [desired update]/[gradient] $\sim \gamma^2$. However, with SignSGD,

the magnitude of the gradient is irrelevant, and we simply want $\eta \sim$ [desired update] $\sim \gamma$. Any smaller learning rate will lead to no training, and any larger learning rate will lead to too large a parameter change, greatly overshooting the target.

In the rich regime ($\gamma \gg 1$), we wish the parameter $w$ to move by an amount scaling as [desired update] $\sim \gamma^{1/L}$. By the same argument, we desire $\eta \sim \gamma^{1/L}$. Unlike for SGD, the max and min viable learning rates are the same in the ultra-rich regime: instead of a finite "triangle of optimizability," the convergent region extends to arbitrarily large $\gamma$. This is because the model escapes the saddle at initialization immediately because the (effectively) small parameters do not suppress each other's gradients. Because of this fast saddle escale, the silent alignment effect does not occur under the SignSGD or Adam optimizers.

It is unknown to the authors whether catapults occur for SignSGD, and if so, whether there is a "catapult triangle" in the phase diagram as seen with cross-entropy loss and SGD Figure 1(b,d).

Results of simulations of our one-parameter toy model with SignSGD are shown in Figure 11.

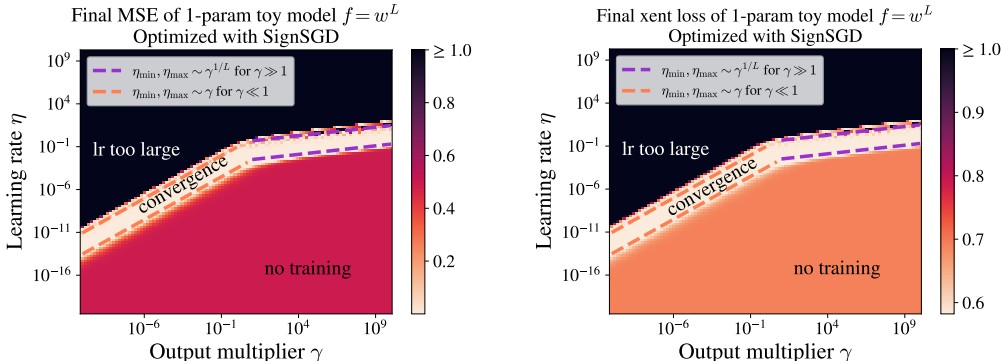

Figure 11: Simulations of our one-parameter model with SignSGD with MSE loss (left; see Equation (4)) and cross-entropy loss (right; see Equation (9)) with varying $\gamma, \eta$ exhibit predicted scaling relationships. Simulations run with $L = 5$. Unlike with SGD, SignSGD does not diverge; instead, points in the upper region oscillate forever.

## F.2 EMPIRICAL PLOTS OF PHASE DIAGRAM ADAM

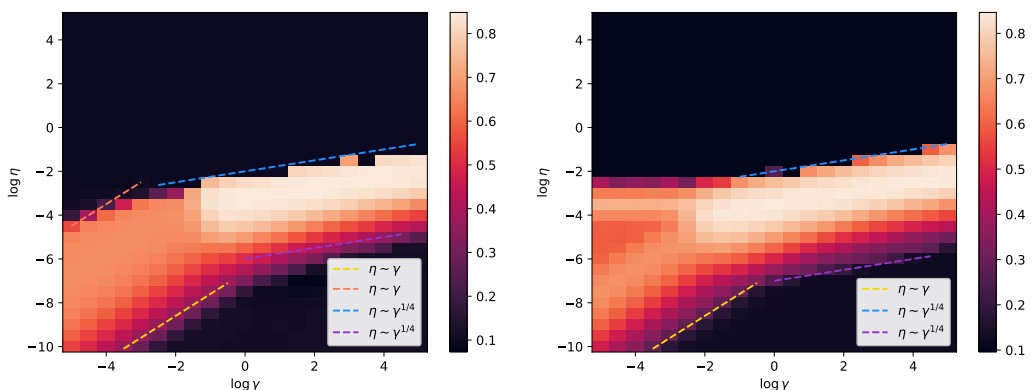

Figure 12: Empirical runs of a 3 layer and readout ConvNet without residuals model trained with Adam with MSE loss (left; see Equation (4)) and cross-entropy loss (right; see Equation (9)) with varying $\gamma, \eta$ on CIFAR-5m.

We supplement the theoretical plots from Figure 11 with empirical plots in a Convolutional Neural Network with 3 layers and a readout layer, no residuals, trained on CIFAR-5m using Adam with

MSE and Cross Entropy losses 12. The models are trained using learning rate warmup for approximately $5\%$ of the total number of steps, followed by a cosine decay until the end of training.

# G ADDITIONAL RESULTS ON SINGLE-INDEX MODELS

In this section we empirically show that the results of the main text transfer to the commonly studied class of "single index models". Here, we train a 3-layer MLP of width 200 with on a single-index model of the form $y = (\boldsymbol{w} \cdot \boldsymbol{x})^k$ where $\boldsymbol{w} \sim \mathcal{N}(0, I)$, $\boldsymbol{x} \sim \mathcal{N}(0, I)$, $\boldsymbol{w} \in \mathbb{R}^D$, $x \in \mathbb{R}^D$. Our setting is the same as in the main text, namely the MLP is parameterized in $\mu$P and is centered. We show the phase portrait of training this model in an online setting with SGD in Figure 13. This portrait is consistent with that predicted by our linear model.

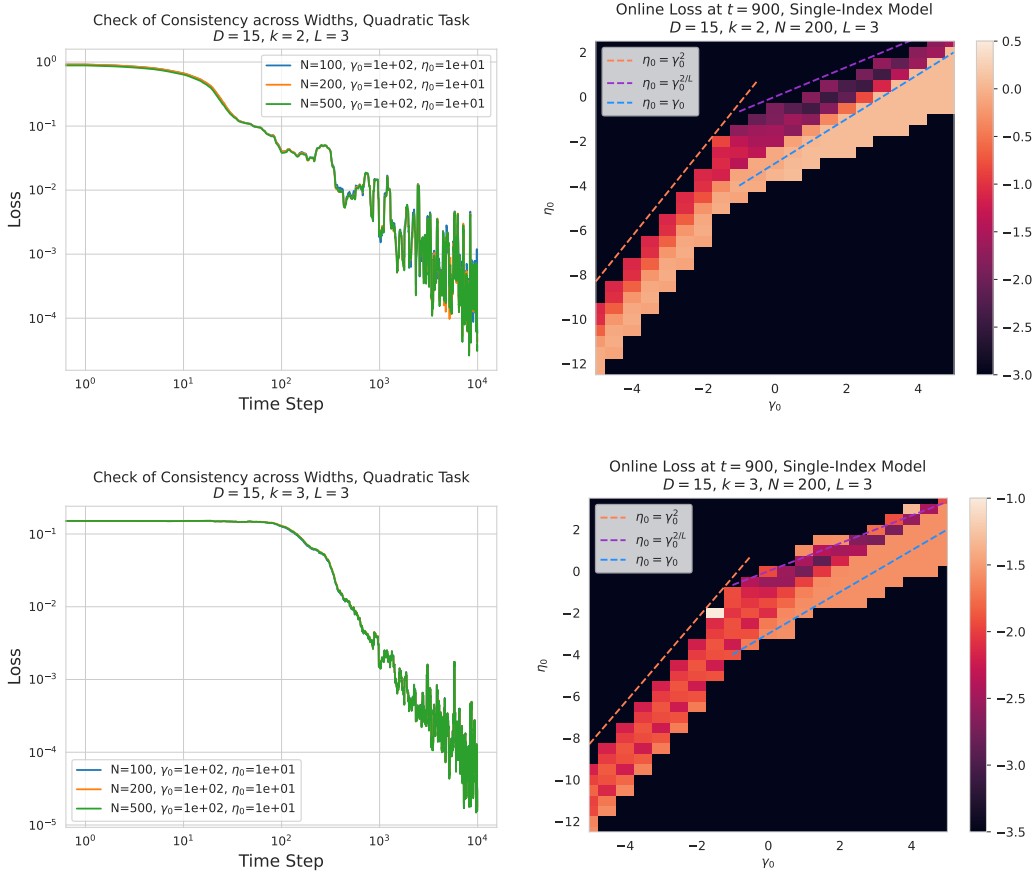

Figure 13: Empirical runs of a 3 layer and readout MLP model trained with SGD and MSE loss on a quadratic single-index model $y = (\boldsymbol{w} \cdot \boldsymbol{x})^2$. a) Verification of consistency across widths in $\mu$P b) The phase plot observed is as in Figure 1. c) and d) illustrate the same, but for $y = 10^{-1}(\boldsymbol{w} \cdot \boldsymbol{x})^3$.

# H ADDITIONAL RESNET AND VIT PLOTS

Here we supplement the $\eta$-$\gamma$ phase plots in the main text with additional plots across different architectures and datasets. We report results for ResNets and visiont transformers. Across all settings, we recover the phase portraits sketched out in Figures 1.

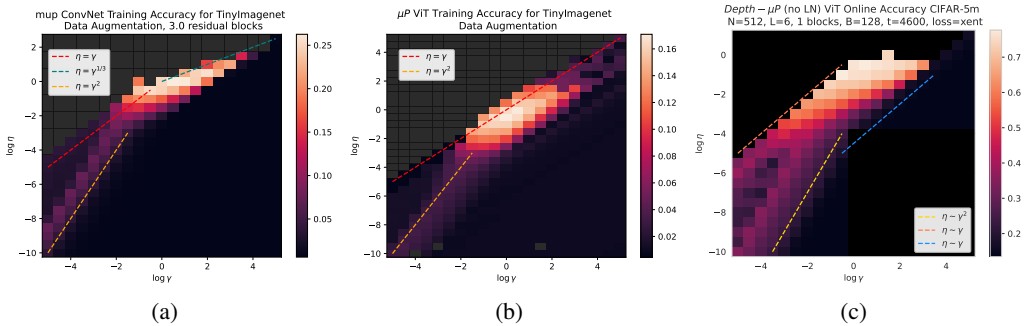

(a)              (b)              (c)

Figure 14: Training accuracy for (a) ConvNets and (b) ViTs (with QK-norm) trained on TinyImagenet, and (c) ViTs without LayerNorm trained on CIFAR-5M for classification using Cross Entropy loss, parameterized with $\mu P$. Removing LayerNorm causes a sharp maximum learning rate cut-off in the training of ViTs. Hyperparameters: batch size $128$, using random data augmentations for TinyImagenet, $1$ head, $3$ transformer blocks.

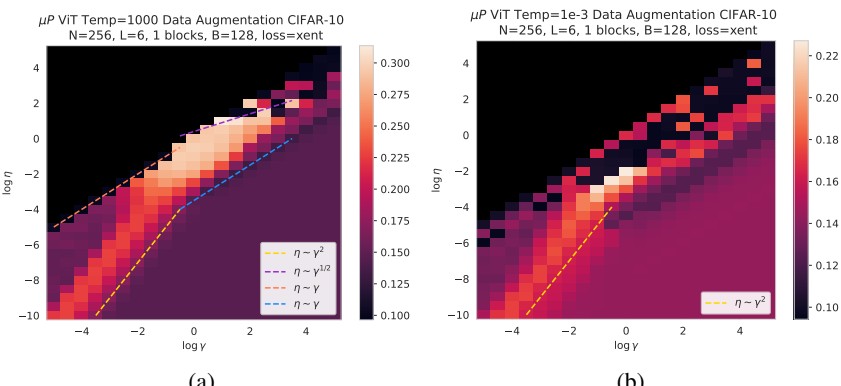

(a)                       (b)

Figure 15: The effect of introducing a temperature $1/T$ parameter in the attention softmax calculation. For large $T$, this makes the attention matrix more uniform which leads to easier optimizability in the large learning rate region. Training is done on CIFAR-10 with random data augmentations, and with LayerNorm.

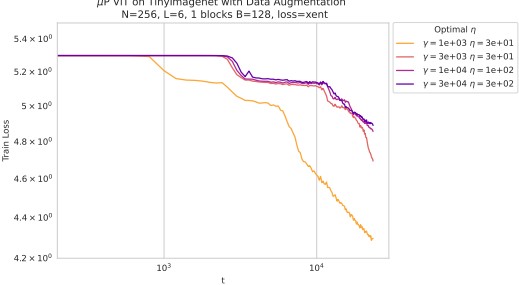

Figure 16: Stepwise loss trajectories can be seen even in the very realistic setting of a vision transformer on TinyImageNet. The trajectories here exhibit two lower plateaus in addition to the long plateau from $t = 0$ until the first drop.

# I  EFFECT OF WIDTH AND CHECK OF $\mu$-CONSISTENCY

In this section we provide checks of $\mu$P working. We do this by both plotting in Figure 17 and Figure 18 as well as their CNN and croseentropy analogues. We observe the following:

- The loss curves as a function of time across widths begin to match as $N$ grows, confirming that they are converging to an "infinite width" loss curve
- The performance vs the learning rate $\eta$ across width is stable, as expected in $\mu$P.

Importantly, as a consequence of consistency across widths at sufficiently large $N$, one does not need to additionally vary the width parameter in order to exhaustively sweep out the space of feature learning NNs.

## I.1  $\gamma$-DEPENDENCY OF $\mu$-CONSISTENCY

Here, we study to what extent width-consistency is violated as a function of $\gamma$.

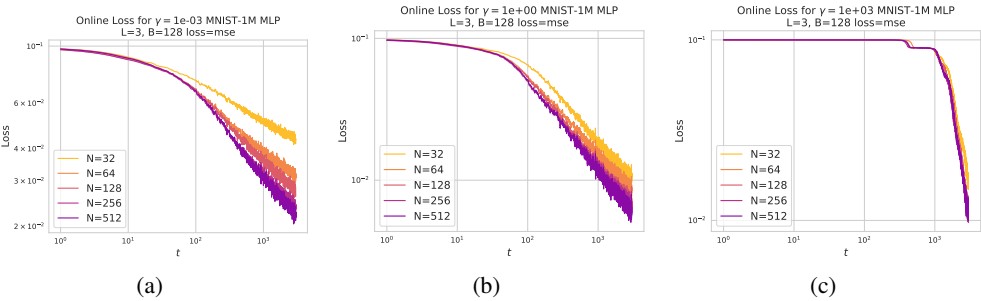

Figure 17: We sweep over $N$ at different values of $\gamma$. We see that the dynamics across different $N$ are closer at larger $\gamma$, indicating they are converging faster to their infinite width feature learning limit than the lazy networks are. We further see that during training we achieve a wider is better effect, as expected from $\mu$P.

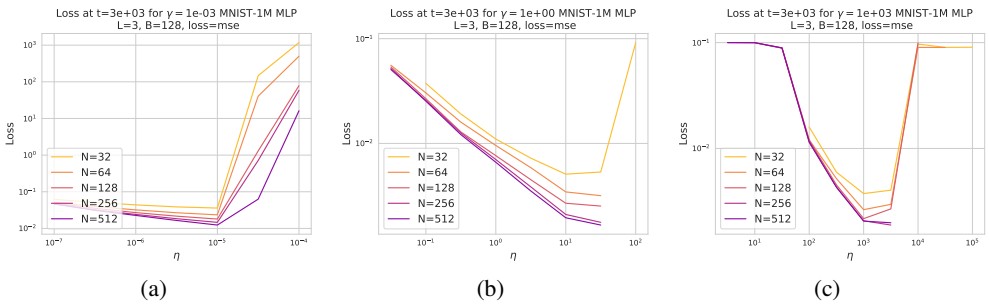

Figure 18: We sweep over $N$ at different values of $\gamma$ and track the optimal $\eta$ across $N$. Plotted is the loss at the end of training. We observe that the minimum essentially remains stable.

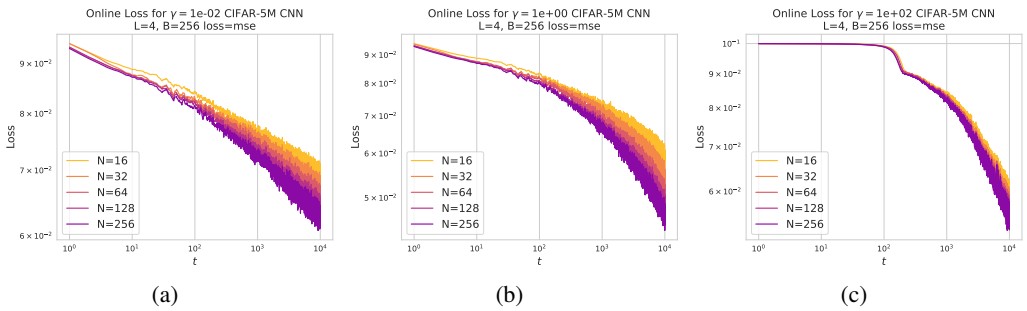

Figure 19: The same as 17, but for a CNN on CIFAR-5M with MSE loss.

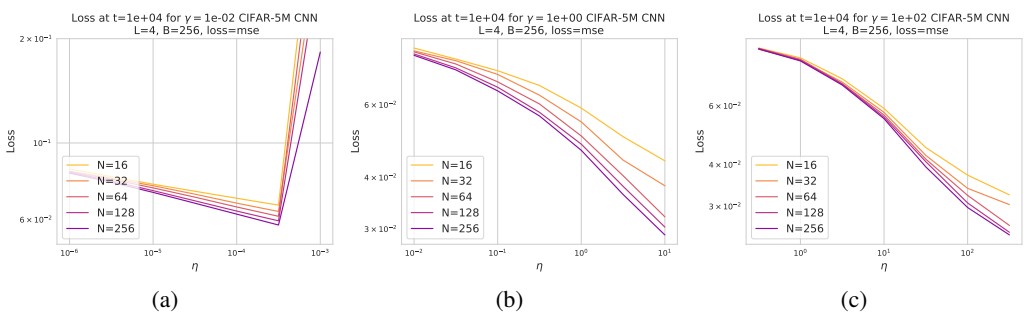

Figure 20: The same as 18, but for a CNN on CIFAR-5M with MSE loss.

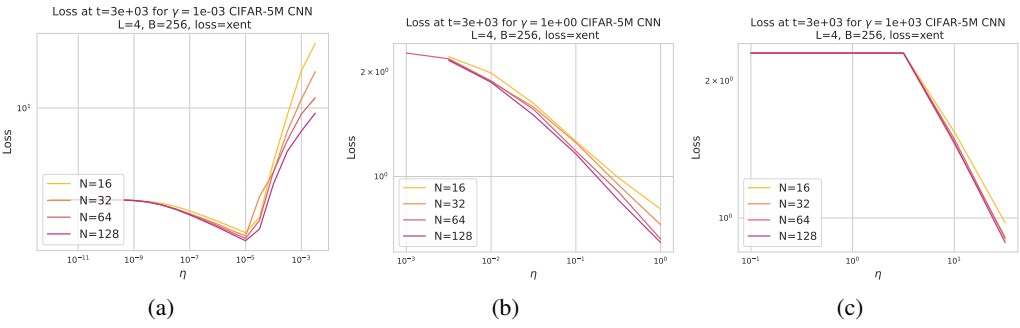

Figure 21: The same as 18, but for a CNN on CIFAR-5M with cross-entropy loss.

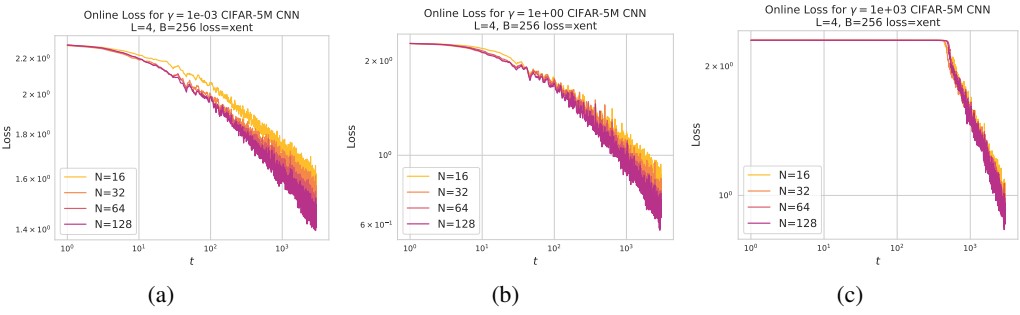

Figure 22: The same as 17, but for a CNN on CIFAR-5M with cross-entropy loss.

# J  EFFECT OF DEPTH

In this section we vary the depth $L$ for both feed-forward MLPs and for ResNets. In the MLP setting, we find a straightforward change in the $\eta - \gamma$ plane. In the rich regime, the maximal observable learning rate scales as $\gamma^{2/L}$, consistent with our model in 4.

## J.1  MLPs

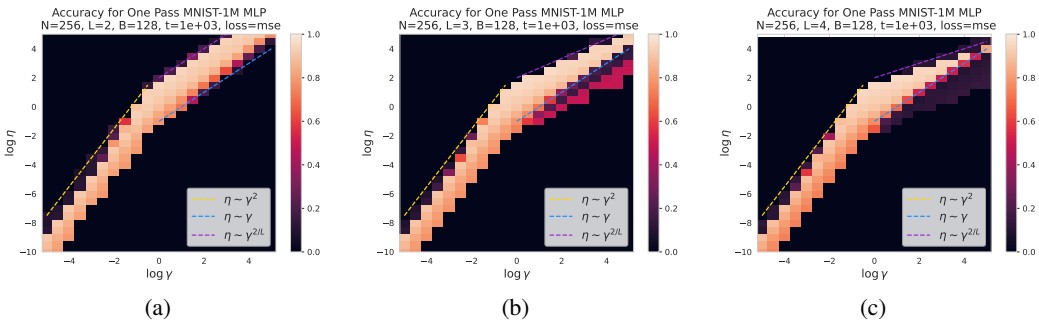

(a)  (b)  (c)

Figure 23: Accuracies after 1000 steps of MLPs trained on MNIST-1M of depth a) $L = 2$, b) $L = 3$, c) $L = 4$. We see that only for the single hidden layer neural network can arbitrarily large $\gamma$ values be optimized. As $L$ grows in the feed-forward setting, the triangle of large $\gamma$ optimizability shrinks.

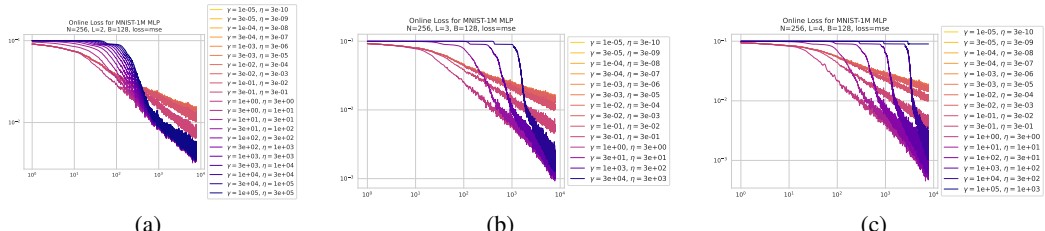

(a)  (b)  (c)

Figure 24: The loss trajectories from selected pixels in Figure 23. We restrict ourselves to the line $\eta = 3\gamma^2$ if $\gamma \leq 1$ and $\eta = 3\gamma^{2/L}$ if $\gamma > 1$ and only plot pixels in the $\eta$-$\gamma$ grid that are within $10\%$ of this line.

## J.2 RESNETS AND DEPTH-$\mu$P

In this section we test the effect of the parameterization on deep neural networks with residual connections. Specifically, in Figure 26 and Figure 25 we evaluate the phase plots of neural networks with increasing depths parameterized in Depth-$\mu$P, as detailed in (Bordelon et al., 2023; Yang & Littwin, 2023), and $\mu$P.

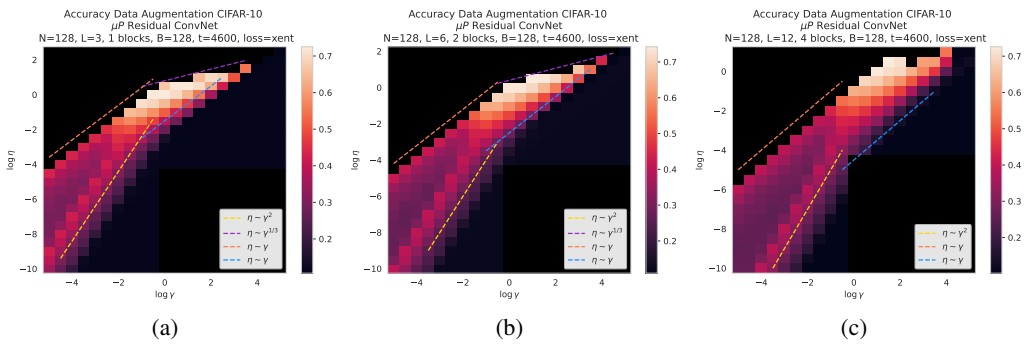

(a)  (b)  (c)

Figure 25: We compare ConvNets with residual connections parameterized in the $\mu$P regime in the online setting by training on CIFAR-5M and testing on CIFAR-10. In this setting the scaling seems to change as we increase the depth of the networks by adding multiple residual blocks. In the very deep setting shown in (c), the very rich networks have a flat $\eta \sim \gamma$ scaling, unlike in the Depth-$\mu$P case where networks across all depths have similar scalings.

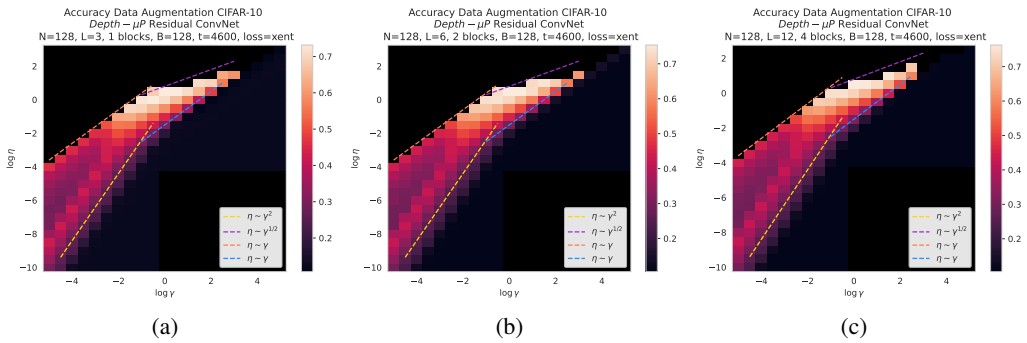

(a)  (b)  (c)

Figure 26: We compare ConvNets with residual connections parameterized in the Depth-$\mu$P regime in the online setting by training on CIFAR-5M and testing on CIFAR-10. Note that the scaling remains much more stable than in Figure 25. We hypothesize that this is due to the downscaling of the residual branch by $\frac{1}{\sqrt{L}}$.

## K  EFFECT OF BATCH SIZE

In this section, we verify that in the online limit, different batch sizes have a relatively negligible effect on the dynamics. The batch noise does however make it so that smaller batches cause higher variance in the loss. This is the tradeoff for being able to train to a lower loss at fixed dataset.

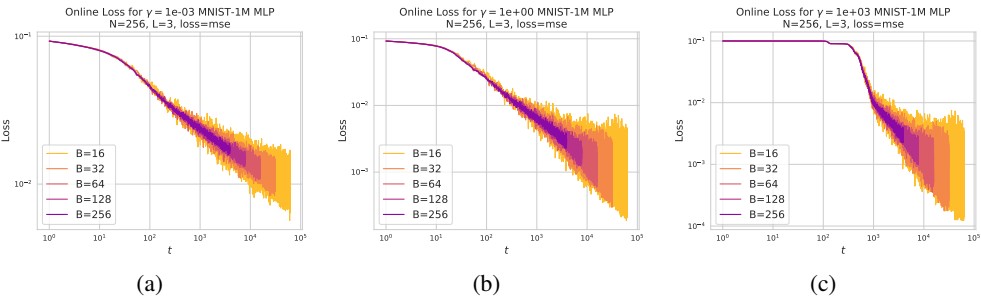

Figure 27: The temporal dynamics of online loss versus time across different values of $B$ at fixed $\eta$. Predictably, smaller $B$ cause larger batch noise variations, but the mean loss path is essentially identical across orders of magnitude in $B$. Upon fixing a data or compute constraint, smaller $B$ networks can train to lower loss values. Larger $B$ networks however are preferable under a wall-clock time constraint.,

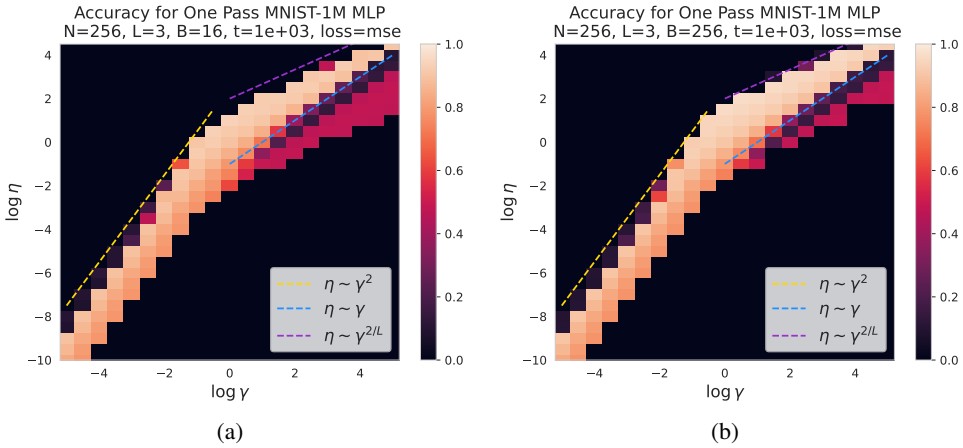

Figure 28: The $\eta$-$\gamma$ phase plot for two MLPs trained for 1000 steps on MNIST-1M for a) $B = 16$ and b) $B = 256$. The location of the dashed lines is *unchanged* from a) to b) . We see that the scalings of $\eta$ with $\gamma$ are identical. The lazy regime observes essentially no change. However, at large $\gamma$, the maximal optimizable $\eta$ is decreased when $B$ shrinks, leading to a more constrained triangle of optimizability at large $\gamma$.

## L  CONSISTENCY ACROSS $\gamma$

In this section we consider how networks at successive $\gamma$ behave in both the lazy regime $\gamma \ll 1$ and in the rich regime $\gamma \gg 1$. In the lazy regime, past a certain threshold $\gamma$, often on the order of $1e-3$, we see that for the datasets and models considered, the dynamics are nearly identical throughout training. This is strongly indicative that these values of $\gamma$ are small enough to

### L.1  LAZY CONSISTENCY

In this section we verify that the lazy limit is indeed reached at $\gamma = 0$. One simple way to do this is to confirm that, conditional on scaling $\eta \sim \gamma^2$, the dynamics agree across $\gamma$ no matter how small $\gamma$ becomes. We plot this in Figure 29.

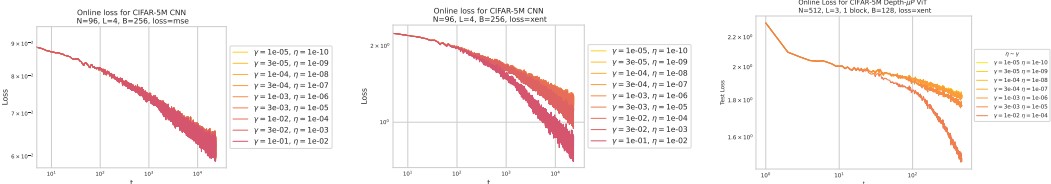

Figure 29: Verification that the lazy limit is reached in practice for CNNs and ViTs trained on CIFAR-5M under MSE and cross-entropy losses. We take $\eta = \gamma^2$ in all plots. Except for a few larger $\gamma$ networks, we see that the loss curves agree very sharply across arbitrarily small values of $\gamma$. The majority of the small $\gamma$ curves are on top of one another in these plots.

### L.2  EARLY TIME RICH CONSISTENCY

In this section, we verify that NNs in the large $\gamma$ limit exhibit similar behavior at early time when time is rescaled to be measured as $\tau = \eta t/\gamma$ when $\eta$ is scaled as $\gamma^{2/L}$. Thus $\tau \sim t\gamma^{-1+2/L}$. This reasoning is explain in Section 4 . We highlight this empirically in Figure 30.

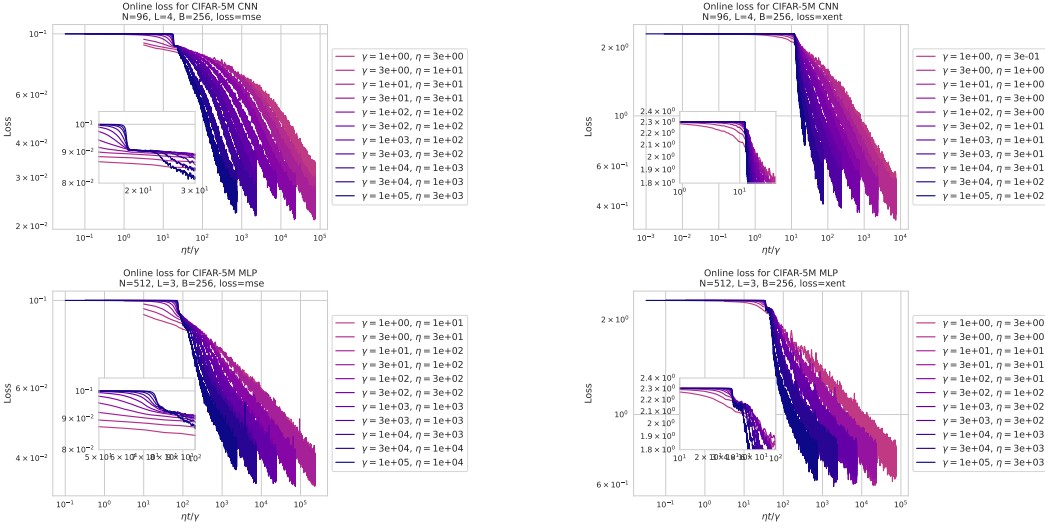

Figure 30: Consistency of dynamics in the large $\gamma$ early time setting for CNNs and MLPs trained on CIFAR-5M. We take $\eta \sim \gamma^{2/L}$ in all plots.

### L.3  LATE TIME RICH CONSISTENCY

We next study the late time dynamics at large $\gamma$. We find that large $\gamma$ networks reach the same loss values as networks with $\gamma \sim O(1)$ after sufficient time passes for them to escape the plateau.

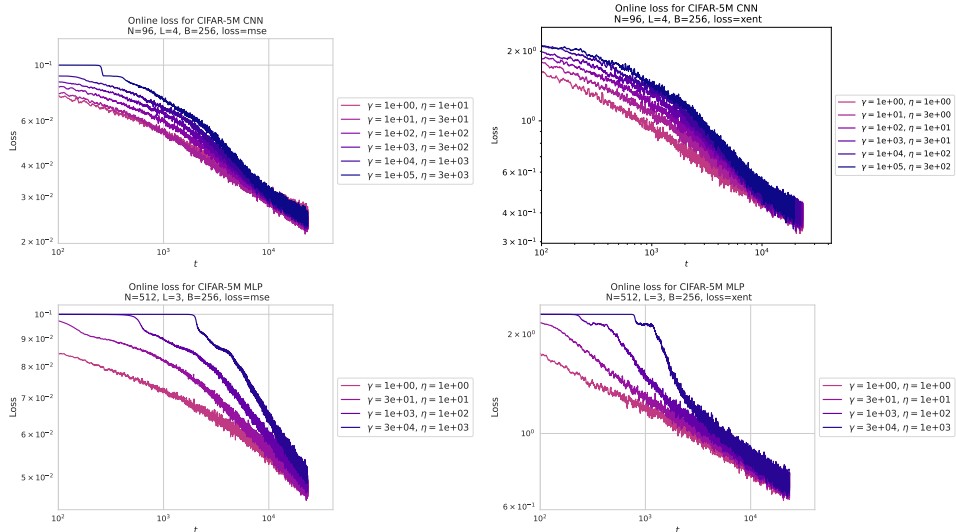

Figure 31: Consistency of dynamics in the large $\gamma$ late time setting for CNNs and MLPs trained on CIFAR-5M. We take $\eta = \gamma^{2/L}$ in all plots. In the inline, we zoom into the location near the first drop, and see that after adopting this time rescaling, all networks achieve the drop at the exact same time.

We also highlight how, after sufficiently long time has passed, all large $\gamma$ networks swept over achieve comparable final losses. One such example was shown in Figure 2(b).

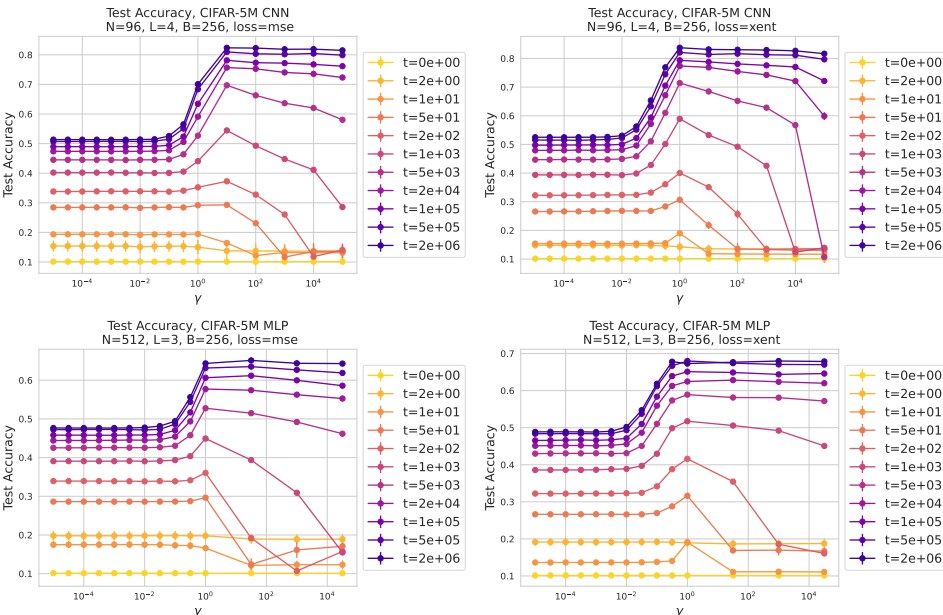

Figure 32: Final accuracies across $\gamma$ for a broader range of networks. We see that for very large $\gamma$ NNs that have not yet escaped the saddle point, the accuracy is not competitive. However, after enough time has passed, these networks also achieve comparable values of the accuracy. The longer optimization time makes such networks undesirable from a practical perspective.

# M  FURTHER HESSIAN PLOTS

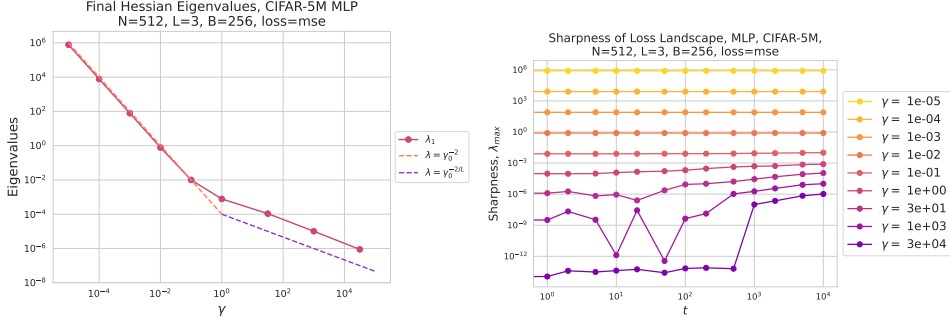

Figure 33:  MLP on CIFAR-5M under MSE loss.  a) The scaling of the sharpness at the end of training as a function of $\gamma$, exhibiting a sharp transition between lazy and rich. b) The sharpness as a function of time across $\gamma$. At large $\gamma$ and early times, calculating the sharpness runs into numerical stability issues that make it unreliable.

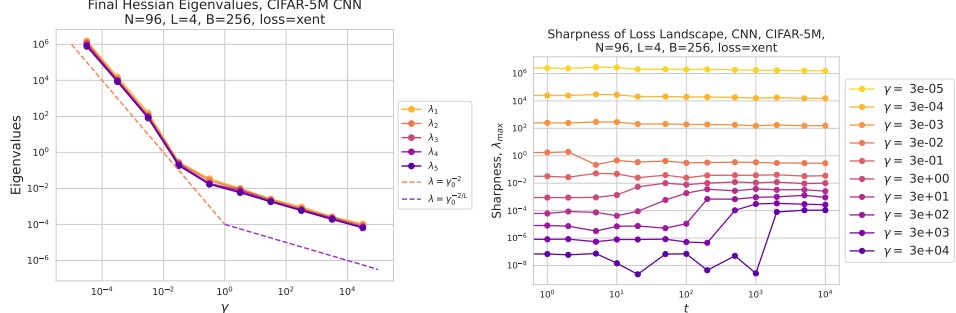

Figure 34:  As in Figure 33, but for a CNN under crossentropy loss, and including the top 5 eigenvalues in a).

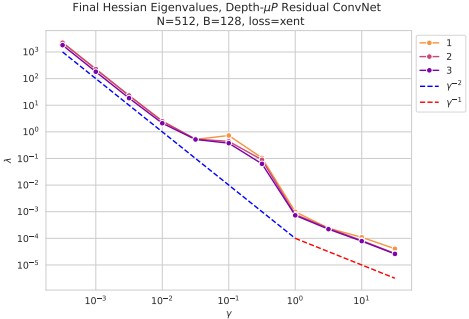

Figure 35:  As in Figure 33, but for a Residual CNN under crossentropy loss, parameterized with Depth-$\mu$P.

# N  FURTHER CATAPULT PLOTS

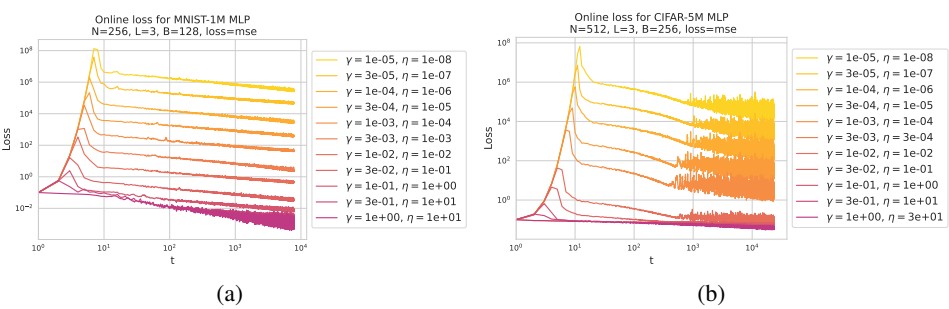

Figure 36: The catapult effect along the $\eta \sim \gamma^2$ line for the MSE loss for MLPs on MNIST-1M and CIFAR-5M

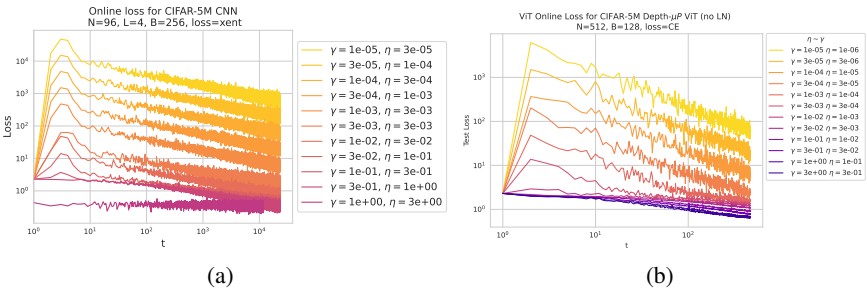

Figure 37: The catapult effect along the $\eta \sim \gamma$ line for the cross-entropy loss for CNNs and ViTs.

# O  ALTERNATIVE SCALINGS OF $\eta$ WITH $\gamma$

In this section, we briefly illustrate how taking alternative scalings of $\eta$ different from $\gamma^{2/L}$ can lead to a network that either undertrains or does not converge at large $\gamma$.

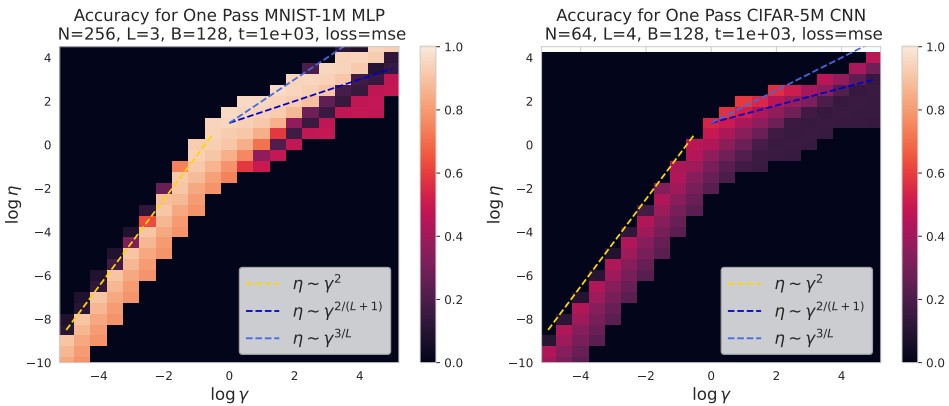

Figure 38: Sketch of phase plots with alternative scalings of $\eta$ with $\gamma$ (overlaid in dashed blue lines). We expect the $\eta \sim \gamma^{3/L}$ scaling to diverge at large $\gamma$ at any $T$ while the $\eta \sim \gamma^{2/(L+1)}$ will remain convergent, but under-train compared to the optimal learning rate

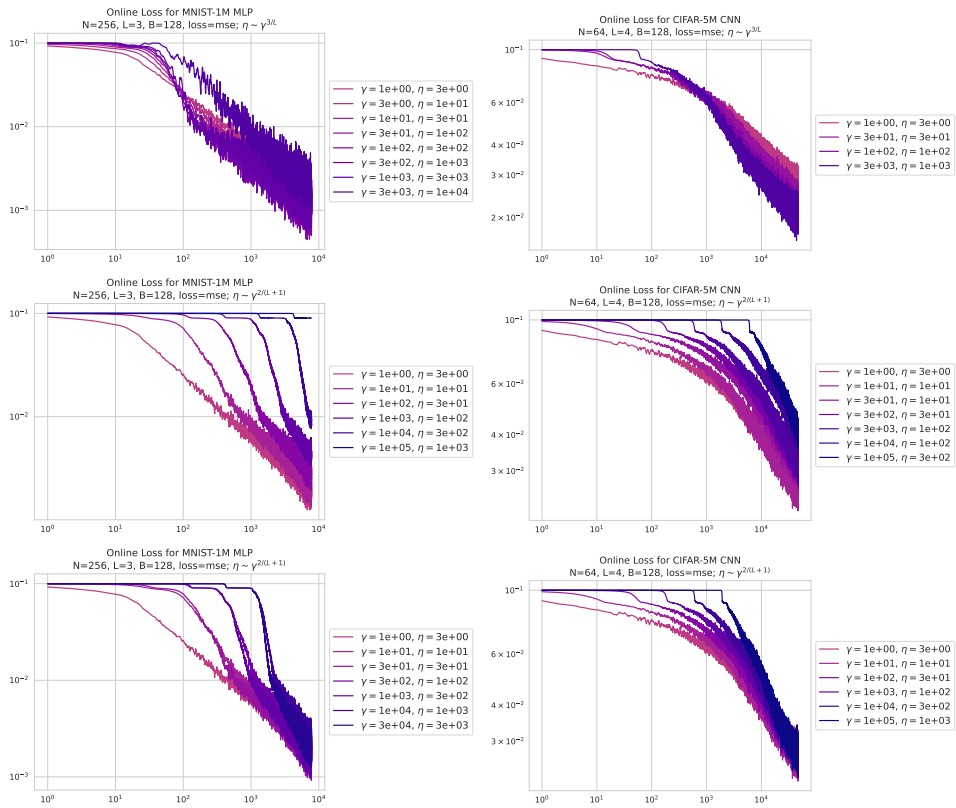

Figure 39: Alternative scalings of $\eta \sim \gamma^{3/L}$ for a) an $L = 3$ MLP on MNIST-1M and b) an $L = 4$ CNN on CIFAR-5M. Large $\gamma$ values not shown are those that have diverged. Indeed, for this scaling, $\gamma$ values above $10^4$ are found to have divergent loss dynamics. Similarly, we plot $\eta \sim \gamma^{2/(L+1)}$ in the same settings in c) and d). All of these networks train, but they do so less efficiently than the optimal scaling of $\eta \sim \gamma^{2/L}$. We illustrate the optimal scaling in figures e) and f), and observe that they achieve better final loss values than c) and d) while still allowing arbitrarily large values of $\gamma$ to train.

## P    KERNEL-TARGET ALIGNMENT

In this section we define what we mean by kernel-target alignment. In the case of a single-class task with labels $y$ for each $\boldsymbol{x}$, we consider a kernel associated to a kernel $K$ originating from an (initialized or trained) neural network. This can be either the neural tangent kernel, $\text{NTK}(\boldsymbol{x}, \boldsymbol{x}') \equiv \nabla f(\boldsymbol{x}) \cdot \nabla f(\boldsymbol{x}')$, or the hidden layer kernel, $K^\ell(\boldsymbol{x}, \boldsymbol{x}') \equiv \phi^\ell(\boldsymbol{x}) \cdot \phi^\ell(\boldsymbol{x}')$. We evaluate $P \times P$ kernel gram matrices $K$ on a held out test set of $P$ points. We stack the corresponding labels into a vector $\boldsymbol{y} \in \mathbb{R}^P$. The associated kernel-target alignment is given by:

$$\text{KTA}(K, \boldsymbol{y}) \equiv \frac{\boldsymbol{y}^\top K \boldsymbol{y}}{\|K\| \|\boldsymbol{y}\|^2}. \tag{24}$$

Here $\|K\|$ is some appropriate norm of the kernel matrix. There are many options, including Frobenius norm, operator norm (maximum singular value), and nuclear norm (sum of singular values). The alignment during the loss plateau is quite sensitive to the choice of norm on $K$,. In practice, we've found that the nuclear norm produces the best results, with Frobenius and Operator norm metrics sometimes resulting in a dip of the alignment before the loss drop rather than a growth.

A related quantity, centered kernel alignment (CKA) (Cortes et al., 2012), has been argued in (Kornblith et al., 2019) to be a good candidate for the comparison of neural representations. In practice, we found direct CKA measures between NTKs of different networks to be very noisy and sensitive. CKA on hidden layer activations yielded more reliable plots.

## Q    FURTHER SILENT ALIGNMENT PLOTS

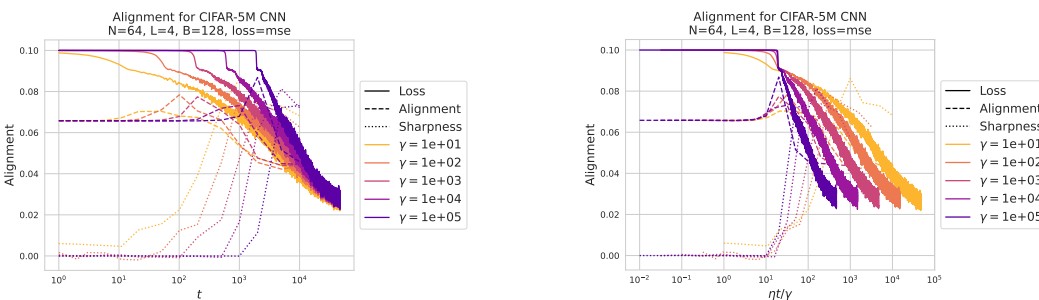

Figure 40: Silent Alignment for a CNN on CIFAR-5M. There is a subtle, but noticeable increase in the alignment metric before the loss drop. a) With $t$ on the x-axis b) with $\eta t/\gamma$ on the x-axis. For a CNN, we see a decrease in the alignment metric after the loss drop. This is due to smaller singular values growing, causing the nuclear norm in the denominator to grow faster than the numerator.

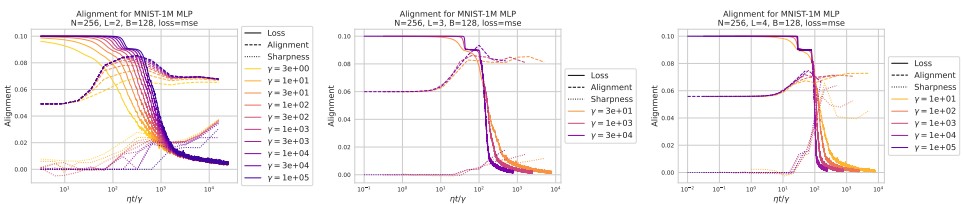

Figure 41: Silent Alignment Across Different Depth MLPs on MNIST-1M

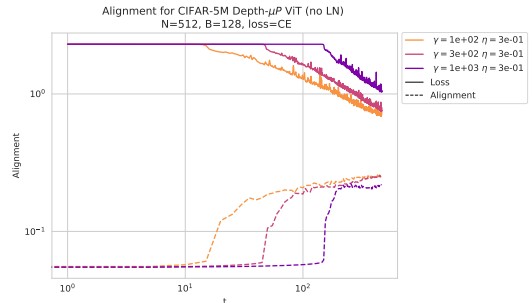

Figure 42: Silent Alignment Across ViTs on CIFAR-5M, evaluated on a single batch of CIFAR-10

# R    FURTHER FUNCTION SIMILARITIES

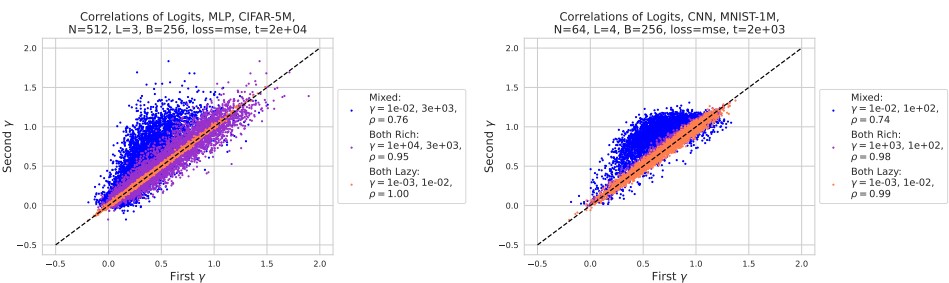

Figure 43: Further comparisons of learned functions between lazy and rich networks.

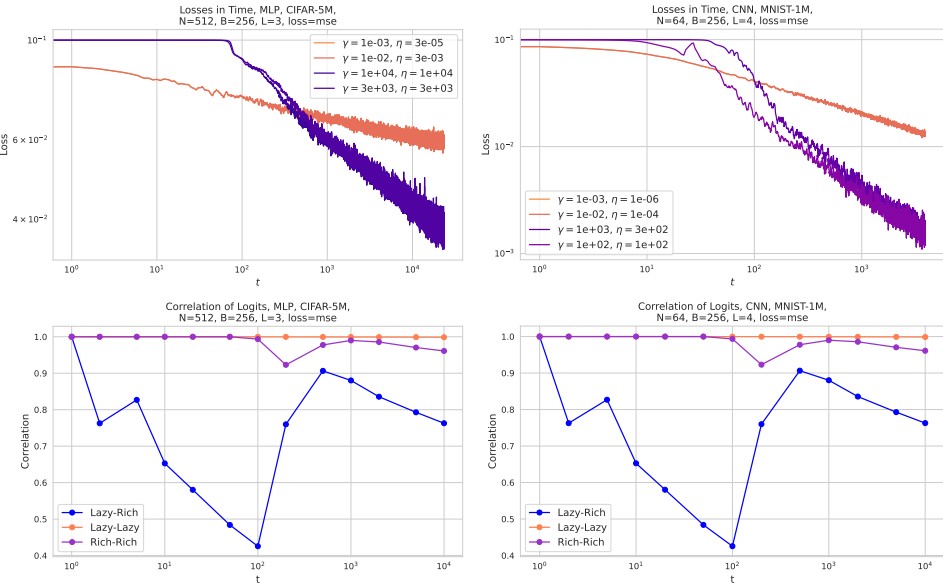

Figure 44: Correlations coefficients between learned functions outputs on a test set between lazy and rich networks in time

# S    WEIGHT AND ACTIVATION MOVEMENT ACROSS TIME AND RICHNESS

In this section, we confirm that larger values of $\gamma$ correlate directly to weight movement across a variety of norms. We consistently verify that the weights deviate as $\|W^\ell(t) - W^\ell(0)\| \sim \gamma^{1/L}$ across layers, times, and $\gamma$.

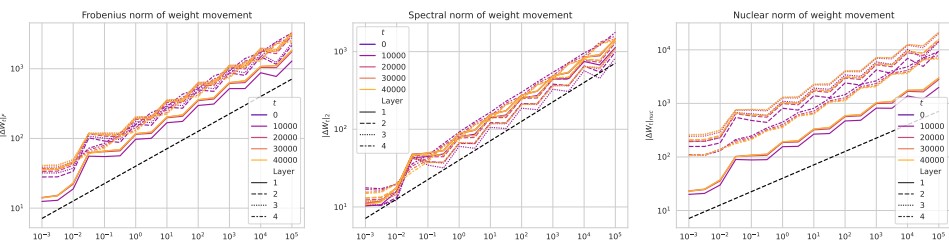

Figure 45: Weight movement $\|W^\ell(t) - W^\ell(0)\|$ for a CNN trained on CIFAR-5M. We plot these across different layers $\ell$ at different steps $t$ of training as a function of $\gamma$. We see a very sharp and clear trend that larger value of $\gamma$ correlate with meaningfully larger movement across a variety of norms. Specifically, we see that all norms of matrices scale as $\gamma^{1/L}$ after sufficient training steps. The dashed black line represents the curve $\gamma^{1/L}$ for reference. a) Frobenius norm, b) Spectral/Operator nom, c) Nuclear norm.

