# OpenReview forum: "The Optimization Landscape of SGD Across the Feature Learning Strength"
_ICLR.cc/2025/Conference — ICLR 2025 Poster_

### Official Review · Reviewer_EZic · 2024-11-01

**Soundness:** 3
**Presentation:** 3
**Contribution:** 2
**Rating:** 6
**Confidence:** 4

**Summary:**

The paper empirically studies the relationships among the scaling hyperparameter $\gamma$, learning rate $\eta$ in neural networks, and the overall performance of neural networks.

**Strengths:**

1. The paper is clearly written and well-organized.
2. The authors conduct a variety of experiments across multiple datasets and network architectures, exploring different combinations of
$(\gamma,\eta)$. The figures presented are particularly effective in conveying the findings.
3. The optimal scaling of $(\gamma,\eta)$ offers valuable insights for selecting hyperparameters in empirical studies.

**Weaknesses:**

1. Although the title emphasizes "Feature Learning", the paper does not adequately elaborate on this concept. There is insufficient empirical evidence regarding whether and how feature learning occurs when $\gamma\gg1$. I recommend that the authors provide more detailed results comparing the weights of networks that exhibit feature learning to those that behave according to the NTK regime.
2. The theoretical framework relies on an oversimplified linear model with a constant target. To gain a better understanding of feature learning, I suggest the authors analyze more complex targets, at least simple single-index models such as $f(x)=\langle a, x \rangle^p$.

**Questions:**

1. In Figure 4, which illustrates the dynamical phenomena under different $\gamma$ settings, the conditions for 4(a) and 4(b) differ (CCN for CIFAR and MLP for MNIST, respectively). Are there similar saddle point phenomena as observed in 4(b) under the conditions of 4(a), or catapult phenomena like those in 4(a) under the conditions of 4(b)?
2. Are there differences in the weights of the first few layers under various $(\gamma,\eta)$ combinations, in addition to the weights of the final layer shown in Figure 5?
3. What are the implications of this work on representation learning?

---

> ### Author Response · Authors · 2024-11-20
> **Response**
>
> We appreciate the reviewer’s kind words regarding our empirical evaluation and on the paper structure.
>
> `Although the title emphasizes "Feature Learning", the paper does not adequately elaborate on this concept. There is insufficient empirical evidence regarding whether and how feature learning occurs when $\gamma >> 1$. I recommend that the authors provide more detailed results comparing the weights of networks that exhibit feature learning to those that behave according to the NTK regime.`
>
> We thank the reviewer for these points.
> 1. Regarding the criticism about empirical evidence on feature learning, we would kindly point the reviewer towards the following figures, which, we believe, provide ample evidence for feature learning:
>
> a. We show in Figure 2 the effect of feature learning across a large sweep of $\gamma$ values, from lazy networks (in the NTK regime) $\gamma = 1e-5$ to “ultra rich” networks $\gamma = 1e5$. Note that the effect of feature learning is present across all the $\gamma > 1$ values since they are able to train much faster than their lazy counterparts.
>
> b. Following, in Figure 3, we show that for lazy networks (in the NTK regime), the Hessian eigenvalues do not evolve from initialization (Figure 3c), whereas in the rich and ultra-rich settings we observe a large deviation from the initialization after training (Figure 3d)
>
> c. Furthermore, In Figure 4b and Figure 5b, we plot the kernel alignment. Note that the lazy networks do not exhibit kernel alignment in the small training step regime, whereas large gamma networks consistently have much larger alignment
>
> d. Finally, we show the correlation of the logits between pairs of networks that are both lazy, both rich, and lazy and rich respectively. Note that between networks with similar levels of “richness” there is a high correlation between the logits, whereas between lazy and rich networks the correlation is very low.
>
> We have thus added Figure 45 illustrating substantial weight movement in the rich, feature-learning regime, and very small weight movement in the lazy limit.
>
> `The theoretical framework relies on an oversimplified linear model with a constant target. To gain a better understanding of feature learning, I suggest the authors analyze more complex targets, at least simple single-index models such as f(x) = \langle a, x \rangle^p`
>
> Finally, we would like to address the comment regarding our theoretical model. Our current theoretical model is able to capture the full $\eta-\gamma$ phase portrait that we have observed empirically across various realistic models and datasets. We would like to comment that the simplicity of our model is indeed a strength and not a drawback. While there may be value in studying interactions with structured target functions like single-index models, that is out of scope for the current work. Nevertheless, we provide empirical plots of learning a simple single-index model in Appendix G, showing that the phase portrait remains unchanged for $p=2$ and $p=3$ single index models.
>
> `In Figure 4, which illustrates the dynamical phenomena under different settings, the conditions for 4(a) and 4(b) differ (CCN for CIFAR and MLP for MNIST, respectively). Are there similar saddle point phenomena as observed in 4(b) under the conditions of 4(a), or catapult phenomena like those in 4(a) under the conditions of 4(b)?`
>
> We thank the reviewer for this question. It’s important that our findings are robust under dataset change. Concretely, you are asking for a) a catapult effect for an MLP on MNIST-1M, and b) a silent alignment saddle point effect for a CNN on CIFAR-5M. The first figure of the appendix section “Further catapult effects” of the revised draft exactly gives a).  For b), we point the reviewer to the first figure of the appendix section titled “further silent alignment plots”. Please let us know if you’d like further plots and we’d be more than happy to generate them.
>
> `Are there differences in the weights of the first few layers under various combinations, in addition to the weights of the final layer shown in Figure 5?`
>
> In order to answer this question we have added Figure 45, which shows that activation (and weight) movement across layers have roughly similar dynamics, under a choice of $\eta$-$\gamma$ pairs.
>
> `What are the implications of this work on representation learning?`
>
> Our work studies the dynamics of neural networks trained in the ultra rich regime. We observe that neural networks at large values of the feature learning parameter seem to learn similar representations (see Appendix Q as well as Fig 39 and Fig 5 left), or at least similar functions late on in training time. We will highlight this more explicitly in the main text. We believe that our work opens up exciting avenues for understanding feature learning in this regime, and we speculate that a further empirical interpretability study into the shape of the learned representations would be a valuable addition to our work.

---

> > ### Comment · Reviewer_EZic · 2024-11-22
> > **Response**
> >
> > Thanks for your response. Although I still believe the theoretical framework is oversimplified, I really appreciate the authors' efforts on the experiments (nearly 50 good figures). Thus, I raised my score to 6.

---

### Official Review · Reviewer_nULc · 2024-11-02

**Soundness:** 3
**Presentation:** 3
**Contribution:** 3
**Rating:** 6
**Confidence:** 3

**Summary:**

The authors conduct a systematic and comprehensive empirical study on the effect of scaling the hyperparameter $\gamma$, located in the final layer, which controls the strength of feature learning, across a variety of datasets and tasks. In this context, a small $\gamma$ corresponds to the kernel regime, while $\gamma = 1$ represents the mean-field regime, where feature learning can still occur even with infinite width.

First, the authors examine the relationship between $\gamma$ and the maximal learning rate, empirically demonstrating that the maximal learning rate scales nontrivially with $\gamma$. Using this optimal learning rate scaling, they then explore the ultra-rich regime, $\gamma \gg 1$. In this large $\gamma$ regime, the network exhibits consistent training dynamics across different values of $\gamma$. Also, the model always achieves optimal performance when $\gamma$ is sufficiently large.

**Strengths:**

1. The experiments are conducted systematically, making the results highly convincing. The paper is also well-written and clear.

2. The observation that networks with a larger $\gamma$ can often achieve the same or even better performance than those with $\gamma=1$ after sufficient training time is intriguing. This suggests a potential new technique to improve performance in practice. It's also interesting to see that networks with large $\gamma$ exhibit very similar training dynamics after an appropriate time rescaling.

**Weaknesses:**

1. I am somewhat unclear on how these results translate to practical applications. While the empirical findings in this paper are robust, the setup does not precisely match real-world scenarios (e.g. the architectures are different). I wonder what insights could be derived if we consider a standard practical setup.

2. I believe the paper would benefit from a more detailed and mathematically formulated introduction to the networks’ structure and parametrization.

**Questions:**

See the weakness part.

---

> ### Author Response · Authors · 2024-11-20
> **Response**
>
> We thank the reviewer for emphasizing the main strengths of our paper. We do agree that given that larger $\gamma$ can obtain better performance than $\gamma=1$  in certain cases makes our work useful for practitioners.
>
> `I am somewhat unclear on how these results translate to practical applications. While the empirical findings in this paper are robust, the setup does not precisely match real-world scenarios (e.g. the architectures are different). I wonder what insights could be derived if we consider a standard practical setup.`
>
> We thank the reviewer for bringing up these 2 very important points regarding our work and we will address them pointwise in the following.
>
> 1. We would like to point the reviewer to two important repositories where a $\gamma > 1$ brings improvements to the accuracy of a classification model trained on CIFAR10: https://github.com/libffcv/ffcv/blob/3a12966b3afe3a81733a732e633317d747bfaac7/examples/cifar/train_cifar.py#L143 or https://github.com/KellerJordan/cifar10-airbench/blob/641e95e360c199cced0b8c1ffdc415f0753921cf/airbench96_faster.py#L50. Furthermore, we would like to kindly point the reviewer to our extensive empirical results on ResNets, and Transformers in Appendix G where we also see that $\gamma > 1$ can bring an improvement (albeit small) in performance. We explain this result intuitively by arguing that empirically, $\gamma=1$ is not “far enough” from the lazy regime in order to see the full benefits of rich training. We also speculate that starting with more uniform logits (i.e. large gamma) is also a better initialization for neural networks.
>
> `I believe the paper would benefit from a more detailed and mathematically formulated introduction to the networks’ structure and parametrization.`
>
> 2. We thank the reviewer for this comment. We agree that it is very important to be explicit about the parameterizations employed. In the original draft, we have elaborated on the difference between parameterizations in Appendix B. Because this is an important aspect of our work, we have updated the main body with a clearer explanation of the network structure, and a more explicit reference to Appendix B, and we have also added a detailed description of the $\mu$P parameterization we have used throughout the manuscript in Appendix A.1.

---

### Official Review · Reviewer_CuAx · 2024-11-03

**Soundness:** 3
**Presentation:** 4
**Contribution:** 2
**Rating:** 5
**Confidence:** 5

**Summary:**

This paper systematically studies the effect of $\gamma$, which is the scale of the output layer of the neural networks, on the optimization and generalization dynamics of neural networks through experiments. They recovered a couple of phenomena that have been analyzed in the literature, including lazying training, catapult effect, silent alignment, etc. They also observed that the generalization performance is similar when $gamma$ is greater than $1$ if the network is trained sufficiently long. Furthermore, they use a simple deep linear neural network with a single univariate data to explain the optimal learning rate scaling.

**Strengths:**

1. This paper is well-written and the logic is easy to follow. The experiments are nicely done and the theoretical results are well presented.

2. There are some interesting empirical observations. For example, they observe that in the feature learning regime i.e., $\gamma \geq 1$, the generalization performance will end up being similar with different $\gamma$, if the learning rate is appropriately chosen. I also find the connection between silent alignment and edge os stability quite interesting and worth further investigation.

**Weaknesses:**

1. One concern is the significance of this paper. Feature learning is an interesting topic, but I believe that this kind of parameterization is not commonly used in practice, i.e., large $\gamma$ with the model being centered. Furthermore, the learning rate used in this paper is constant hence is far from practice. Therefore, I am not sure if the insights observed in this paper can shed light on practical network training.

2. This paper may provide valuable insights for theoretical analysis. However, many of the phenomena discussed, such as lazy training, the catapult phase, and silent alignment, have already been extensively studied. Although the ultra-rich regime is analyzed in depth, it does not appear particularly interesting, as the generalization performance turns out to be similar to that of $\gamma=1$. All in all, the paper presents a nice summary of the network training dynamics, but I do not find anything particularly new or significant.

**Questions:**

1. I am curious how do you conclude that the optimal learning rate is $\gamma^{2/L}$ for $L$-th layer neural network? I understand that for the deep linear network with a single univariate sample the optimal one is that value, but I am surprised that for deep nonlinear networks it still holds, and the power is quite specific. Does the generalization performance have a big gap when the learning rate is $3/L$ or $2/(L+1)$?

2. In practice, the learning rate decay is useful to improve the generalization performance, and it may avoid optimization from divergence. Do you think the optimal learning rate will be different if you include learning rate decay?

---

> ### Author Response · Authors · 2024-11-20
> **Response - part 1**
>
> We thank the reviewer for their kind words on our work.
>
> We understand the concerns of the reviewer regarding the significance of our paper and we address them pointwise:
>
> `One concern is the significance of this paper. Feature learning is an interesting topic, but I believe that this kind of parameterization is not commonly used in practice, i.e., large with the model being centered. Furthermore, the learning rate used in this paper is constant hence is far from practice. Therefore, I am not sure if the insights observed in this paper can shed light on practical network training.`
>
> 1a) Regarding centering, note that with neural networks parameterized under muP, centering is actually common (and even strongly recommended, see for example Appendix D.2 in [3]). This is done in finite width networks in order to remove the finite width noise and bring the neural networks closer to their infinite width limits. In practice, centering is usually achieved by zeroing out the readout layer at init. This works in the rich and ultra-rich regimes, but it doesn’t make sense in the lazy regime, as training gets concentrated in the last layer. For consistency across regimes, we thus adopted the explicit centering method of subtracting off the function value at init. This is just a methodological choice and we expect it does not affect our findings in the rich and ultra-rich regimes.
>
> 1b) Regarding large $\gamma$, it’s true that most experimentalists implicitly use the naive setting $\gamma = 1$. However, our findings consistently show that the optimal $\gamma$ is often a value greater than one (typically $\gamma \in [5, 10]$). This constitutes a key contribution of our work: we recommend that practitioners explicitly include and tune the hyperparameter $\gamma$. Supporting this recommendation, several high-profile, hyperoptimized public ML benchmarks [1, 2] have independently observed that using output multipliers corresponding to $\gamma$ in this range is beneficial. We present the first theoretical explanation for this observation. We provide the first theoretical explanation for this phenomenon, offering valuable guidance for practitioners. We acknowledge that this point may not have been sufficiently emphasized previously and have revised the draft to address this.
>
> 1c) We generally prefer to use a constant learning rate rather than a schedule. This approach simplifies experimentation and reduces the complexity associated with tuning hyperparameters, allowing for more controlled and reproducible scientific investigations. A follow-up paper aiming to achieve state-of-the-art results on various benchmarks would likely involve tuning both $\gamma$ and the learning rate. However, our focus here is to clearly observe and disentangle dynamical phenomena, where incorporating a learning rate schedule could introduce confounding effects. From a theoretical perspective, the learning rate schedule effectively serves as an order-one scalar multiplier to the learning rate at all times. Consequently, our theoretical results remain unchanged. In order to address this point, we have included Figures E.10 and F.12, which are trained with a 5% warmup followed by a cosine decay schedule, showing that these results do not change the conclusion of our findings.

---

> > ### Author Response · Authors · 2024-11-20
> > **Response - part 3**
> >
> > `I am curious how do you conclude that the optimal learning rate is $\eta^{2/L}$ for $L$-th layer neural network? I understand that for the deep linear network with a single univariate sample the optimal one is that value, but I am surprised that for deep nonlinear networks it still holds, and the power is quite specific. Does the generalization performance have a big gap when the learning rate is $3/L$ or $2/(L+1)$?`
> >
> >
> > Thank you for these questions.
> >
> > 1) We agree it is remarkable that such a simple toy model predicts the correct exponents! This is because the scaling phenomenon here is fundamentally about the sizes (and more specifically, the spectral norms) of the model layers. Our toy model is designed to capture that aspect of deep neural nets; it’s the simplest model that has this property.
> >
> > We have added Appendix O addressing this question as to whether alternative scalings of $\eta$ with $\gamma$ going as 3/L or 2/(L+1) would make a large difference. We attach a link to that figure for an L=3 MLP on MNIST-1M and an L=4 CNN on CIFAR-5M. In both cases, we see that for even moderately large $\gamma$ the 3/L scaling quickly diverges in training time. This is also clear from the central phase diagram. By contrast, the 2/(L+1) scaling is indeed always convergent, but underperforms the 2/L scaling due to having a sub-optimal learning rate.
> >
> > `In practice, the learning rate decay is useful to improve the generalization performance, and it may avoid optimization from divergence. Do you think the optimal learning rate will be different if you include learning rate decay?`
> >
> > 2) Regarding learning rate decay, as long as the lr schedule takes the form of an order-unity multiplier like $\text{[lr]} = s(t) \cdot \eta$, with $s(t)$ independent of $n, \eta$, our scaling analysis remains the same.
> > Theoretically, this occurs because we assume that our scaling parameters, namely the $\gamma$ parameter, is large enough that it dominates any constant factor change brought up by the scheduler. In practice, we expect that adding a learning rate schedule would keep our phase diagram but slightly change the shape of the “transition region” around $\gamma \approx 1$. We confirm this hypothesis in Appendix E and F.2 which we have now included, showing the phase diagram under SGD and Adam with warm up followed by learning rate decay. In the case of SGD, the shape of the phase diagram is unchanged from the constant learning rate setting.

---

> ### Author Response · Authors · 2024-11-20
> **Response - part 2**
>
> `This paper may provide valuable insights for theoretical analysis. However, many of the phenomena discussed, such as lazy training, the catapult phase, and silent alignment, have already been extensively studied. Although the ultra-rich regime is analyzed in depth, it does not appear particularly interesting, as the generalization performance turns out to be similar to that of. All in all, the paper presents a nice summary of the network training dynamics, but I do not find anything particularly new or significant.`
>
> 2) The primary novelty and significance of our theoretical analysis lie in the exploration of the large-$\gamma$ limit. While the study of the lazy ($\gamma \ll 1$) limit and kernel dynamics has been foundational to deep learning theory, our empirical evidence points to the existence of an opposing limit with distinct phenomena and without the NTK's shortcomings, such as the loss of feature learning and reduced performance. This work also provides a strong recommendation to treat $\gamma$ as a tunable parameter, alongside a thorough characterization of the associated training behaviors, offering valuable insights for both theory and practice.
>
> Many prior works, including the catapult effect paper, have studied the dependence of the phenomenon on network width. In $\mu$-parameterization, however, the dynamics remain consistent across widths, rendering width a less critical factor in these phenomena. Our study is the first to empirically demonstrate in realistic settings that the catapult effect can emerge at arbitrarily large widths, driven solely by small $\gamma$ values. Bordelon et al. [https://arxiv.org/pdf/2304.03408] notes this can happen in a perturbative expansion on toy data Our work is, to our knowledge, also one of the first to show that silent alignment can hold for a much broader range of empirical conditions than those studied in the original paper. Lastly, we have updated our rebuttal version with Appendix E, showing theoretical predictions in the simplified model for SignSGD, and empirical runs with Adam showing that the toy model is able to capture almost perfectly the true phenomenon.
>
> [1]: Link: [https://github.com/libffcv/ffcv/blob/3a12966b3afe3a81733a732e633317d747bfaac7/examples/cifar/train_cifar.py#L143]
>
> [2]: Link: [https://github.com/KellerJordan/cifar10-airbench/blob/641e95e360c199cced0b8c1ffdc415f0753921cf/airbench96_faster.py#L50]
>
> [3]: Yang, Greg, et al. "Tensor programs v: Tuning large neural networks via zero-shot hyperparameter transfer." arXiv preprint arXiv:2203.03466 (2022).

---

> ### Comment · Reviewer_CuAx · 2024-11-23
>
> I thank the authors for the response. I have a few more comments.
>
> > Supporting this recommendation, several high-profile, hyperoptimized public ML benchmarks [1, 2] have independently observed that using output multipliers corresponding to $\gamma$ in this range is beneficial.
>
> Thank you for providing the references. It seems like they are using a small $\gamma$. For example, they multiply the last layer weights by $0.2$ in [1] and $1/9$ in [2]. Does it contradict your large $\gamma$ regime? Please correct me if I misunderstand it.
>
> > We provide the first theoretical explanation for this phenomenon, offering valuable guidance for practitioners.
>
> I have seen such feature learning analysis with large $\gamma$ in literature, for example [Woodworth 2020]. I guess to the best of my knowledge you are the first to show that the optimal $\gamma$ is between $5$ and $10$. But I don't see why it is particularly interesting for practitioners since your results are not state of the art and it is unclear if it can improve the performance in practice since there are so many tunable hyperparameters.
>
> Woodworth, Blake, et al. "Kernel and rich regimes in overparametrized models." Conference on Learning Theory. PMLR, 2020.
>
> I still feel that this paper presents a few interesting findings, but none of them are sufficiently significant or novel. I do not believe that a collection of small contributions will amount to a significant one. Therefore, I am inclined to maintain my score.

---

> ### Author Response · Authors · 2024-11-23
> **Reply**
>
> We thank the reviewer for their reply and we offer several clarifications below.
>
> > Thank you for providing the references. It seems like they are using a small $\gamma$. For example, they multiply the last layer weights by $0.2$ in [1] and $1/9$ in [2]. Does it contradict your large regime? Please correct me if I misunderstand it.```
>
> We thank the reviewer for this question and we are happy to provide details. The values the reviewer is pointing to are indeed consistent with ours. Similar to previous works [1], $\gamma$ is the value in the denominator (i.e. the neural network is multiplied by $1/\gamma$), hence in the provided examples, $\gamma=5$ and $\gamma=9$ respectively. We provide details on the exact structure of the neural network and the $\gamma$ factor in Eq. 1 and Appendix A.
>
> > I have seen such feature learning analysis with large in literature, for example [Woodworth 2020]. I guess to the best of my knowledge you are the first to show that the optimal $\gamma$ is between $5$ and $10. But I don't see why it is particularly interesting for practitioners since your results are not state of the art and it is unclear if it can improve the performance in practice since there are so many tunable hyperparameters.
>
> We thank the reviewer for this comment. First, we would like to point out that the references work by Woodworth et al. does not do any learning rate analysis, and only focuses on the interplay between width, depth and the $\gamma$ value. In contrast, we study the relationship between the learning rate and $\gamma$ in our work. Furthermore, our work is in the online setting and in $\mu$P, whereas Woodworth et al. do multiple passes over the data. Finally, we are able to characterize analytically the empirical phase portrait we observe across multiple realistic architectures, optimizers and datasets.
>
> Moreover, while our results are not state of the art, we do show empirically that sweeping over $\gamma$ does indeed bring improvements to a model's performance. Therefore, we argue that the richness is an important parameter to be sweeped over by practitioners and further work on the more fine-grained effect of this hyperparameter on models such as Transformers is an interesting avenue: for example, different $\gamma_l$ on every residual branch, the effect of $\gamma$ on self-attention.
>
> [1]: Bordelon, Blake, and Cengiz Pehlevan. "Self-consistent dynamical field theory of kernel evolution in wide neural networks." Advances in Neural Information Processing Systems 35 (2022): 32240-32256.

---

### Official Review · Reviewer_aPNz · 2024-11-03

**Soundness:** 3
**Presentation:** 2
**Contribution:** 3
**Rating:** 8
**Confidence:** 3

**Summary:**

The paper studies the scaling law between $\gamma$, the feature learning strength parameter, and $\eta$, the learning rate, both empirically in a diverse set of data (MNIST, CIFAR, TinyImageNet) and model architectures (MLPs, CNNs, Vision-Transformers), and theoretically in simple setting with deep linear network. They observed a characteristic phase portrait in the $\gamma-\eta$ plane, and showed that the learning rate scales as $\eta \sim \gamma^2$ in the lazy regime ($\gamma << 1$) and $\eta \sim \gamma^{2/L}$ in the rich regime ($\gamma >> 1$). They confirmed this phase portrait empirically by sweeping over every pair of $\gamma, \eta$ in a log-spaced grid running from $\gamma = 10^{-5}$ to $10^5$ and analytically by theoretically analyzing a deep linear network models for both MSE loss and cross-entropy loss. They also observed and reported various dynamical phenomenon such as "catapult effect", "silent alignment", and "progressive sharpening".

**Strengths:**

The paper provides extensive empirical study for the scaling law of $\eta$ (learning rate) and $\gamma$ (feature learning strength), and also provided an analytical explanation using deep linear network that matches with the empirical results.

**Originality**
1. The paper investigates the scaling relationship between feature learning strength $\gamma$ and learning rate $\eta$ in both lazy, rich, and ultra-rich regime. To my knowledge, I haven't seen work that extensively address this topic.

**Quality**
1. Empirical study: The paper provided an extensive empirical study across $\gamma$ and $\eta$ plane, ranging from $\gamma = 10^{-5}$ to $10^5$ and $\eta = 10^{12}$ to $10^{-12}$ with a diverse set of dataset (MNIST, CIFAR, TinyImageNet) and model architectures (MLP, CNN, Vision Transformer), and confirmed that the claimed scaling laws are observed across different datasets and models.
2. Theoretical analysis: In a simple settings (deep linear network), the paper derived the $\gamma-\eta$ scaling law for both MSE and cross-entropy loss, and showed that the theoretical results match with the empirical observations. The authors also produced an analytical phase portraits in Fig 6 that match with the empirical phase portraits from parameters sweeping.
3. Analysis of dynamical phenomenon: The paper also reported various learning dynamic phenomenon that was reported in earlier works, including catapult effect, silent alignment, step-wise loss drops, progressive sharpening, in various $\gamma-\eta$ region that they perform the  parameter sweeping.

**Significance**
1. The paper showed the scaling relationship between feature learning strength $\gamma$ and learning rate $\eta$ in both lazy, rich, and ultra-rich regime. This scaling law enables the authors to find the range of possible learning rate to train models in ultra-rich regime, an area that few previous work touched on. Using the suitable learning rate, the authors show that in models trained in the ultra-rich regime can exceed or match the performance of models trained in rich regime, which open a potential area to improve current model training dynamics.

**Weaknesses:**

While the paper addressed a significant topic in the feature learning area and provided reasonable empirical and theoretical results, I think the clarity of the paper still have lots of room for improvement to help the readers have an easier time to read the paper. Specifically, the authors can consider to try to re-organize the paper flow to focus on the key points: Since the key results of the paper is the scaling relationship between feature learning strength $\gamma$ and learning rate $\eta$, the paper can be reorganize so that these key theoretical and empirical results come first and emphasized in the paper, instead of leaving the theoretical results at the end of the paper, and mention many other detailed results in the middle of the paper.

**Questions:**

**Questions**
1. [L137] Why further downscale by $1/\gamma$? Isn’t the $\mu P$ scaling already scale by a factor of $1/\gamma$?
2. I couldn't find any information in the paper about how many repetition were run for each empirical results? Would appreciate if you can provide some information about this.

**Suggestion**
1. [L068] Key contribution should be clear and emphasizing the key points (usually 3-4 key points). This current key contribution section is too long and difficult to follow. I’d appreciate if the authors can re-organize this section to make the key contribution clearer to the readers.
2. [L068] Hyperlink each key contribution with the section and/or figure that demonstrate the key contribution so that readers have easier time to follow.
3. [L215] Explicitly mention whether this scaling results are for MSE or cross-entropy loss, since they have different scaling. I was quite confused when the author states that “At small $\gamma$, $\eta \sim \gamma^2$ for lazy networks”, but Fig 2d shows that $\eta \sim \gamma$ instead, but later I realized that the claim “At small $\gamma$, $\eta \sim \gamma^2$ for lazy networks” is for MSE, and the authors have another section for cross-entropy loss.

---

> ### Author Response · Authors · 2024-11-20
> **Response**
>
> We thank the reviewer for the thorough evaluation of our paper and for recognizing the significance of our work. Furthermore, we appreciate the reviewer’s excellent advice on the reorganization of our paper towards emphasizing the theoretical results first, followed by the empirics. We will apply these changes in full in the next paper revision.
>
> `[L137] Why further downscale by $1/\gamma$? Isn’t the $\mu$P scaling already scale by a factor of $1/\gamma$?`
>
> 1. We thank the reviewer for this observation. This is an important subtlety. In prior works [1], the \gamma factor is introduced and taken to scale as $\gamma_0 \sqrt N$ while the learning rate is taken to scale as $\eta_0 \gamma_0^2 N$. Importantly, even in that work there is a residual $\gamma_0$ parameter that controls the feature-learning strength once in $\mu$P.
> In our notation, our $\gamma$ corresponds to the $\gamma_0$ in bordelon et al, and our $\eta$ corresponds to $\eta_0 \gamma_0^2$ in that work. We find this to be a more straightforward notation that avoids the $_0$ subscripts. To avoid confusion, we have added a section in the appendix clarifying our notation.
>
> `I couldn't find any information in the paper about how many repetition were run for each empirical results? Would appreciate if you can provide some information about this.`
>
> 2. We apologize for not being more explicit about our repetitions. Although we did mention in the appendix that the MNIST-1M results were performed online (no data repetition) and run over three different initialization seeds to obtain error bars, we have extended this to clarify that the same holds for CIFAR-5M. For TinyImageNet results and the runs over ResNets and ViTs, we have a single seed, due to compute limitations.
>
> Finally, we thank the reviewer again for their suggestions regarding the reorganization of our results, as well as their suggestions regarding linking the key contributions to their respective sections, and to be more detailed regarding the loss we use in each scaling plot. We will address these comments and include the changes in the revised version of the paper.
>
> [1] Bordelon, Blake, and Cengiz Pehlevan. "Self-consistent dynamical field theory of kernel evolution in wide neural networks." Advances in Neural Information Processing Systems 35 (2022): 32240-32256.

---

> > ### Comment · Reviewer_aPNz · 2024-11-26
> > **Response**
> >
> > Thank the authors for addressing my questions! I've decided to keep my initial score and recommend "Accept" for this paper.

---

### Meta-Review · Area_Chair_ophr · 2024-12-20

**Metareview:**

The paper “The Optimization Landscape of SGD Across the Feature Learning Strength” investigates the interaction of the scaling hyperparameter
𝛾
γ, which controls the strength of feature learning, and the learning rate
𝜂
η across various datasets, models, and training regimes. The authors provide a comprehensive empirical analysis complemented by theoretical insights derived from a deep linear network. Notably, the paper explores the less-studied "ultra-rich" feature learning regime, offering novel scaling relationships and dynamical phenomena such as the catapult effect and silent alignment. The authors also identify optimal learning rate scaling laws that vary across regimes, validated by experiments spanning MLPs, CNNs, Vision Transformers, and datasets such as MNIST, CIFAR, and TinyImageNet.

Strengths:

Comprehensive Empirical Study: The paper presents a robust empirical evaluation, covering a broad parameter space (
𝛾
γ and
𝜂
η) across multiple architectures and datasets.
Novel Insights: The exploration of the ultra-rich regime and the scaling relationship between
𝛾
γ and
𝜂
η adds significant theoretical and practical insights to the field.
Strong Theoretical Component: The analytical results, derived from a simplified deep linear network, align closely with empirical observations, providing a solid theoretical grounding for the findings.
Presentation and Clarity: The paper is well-organized, with effective use of figures to communicate results.
Weaknesses:

Practical Relevance: While the paper’s findings are theoretically insightful, their applicability to standard training setups with learning rate decay and more complex architectures requires further elaboration.
Clarity of Key Contributions: Reviewers noted that the presentation could benefit from reorganization to highlight the central theoretical contributions more explicitly.
Simplified Model Assumptions: The theoretical model, while effective, may not fully capture the complexities of real-world feature learning scenarios.
Limited Empirical Validation in State-of-the-Art Contexts: While the paper provides valuable insights, it does not demonstrate direct improvements in state-of-the-art performance benchmarks.

**Additional Comments On Reviewer Discussion:**

The paper provides substantial theoretical and empirical contributions to understanding the optimization landscape of neural networks under varying feature learning strengths. The novelty of exploring the ultra-rich regime and the scaling laws for  𝛾 γ and 𝜂 η make it a valuable addition to the field. Despite some limitations in practical applicability and the simplicity of the theoretical model, the paper’s strengths outweigh its weaknesses. The empirical results are robust, and the theoretical insights provide a strong foundation for future work. Therefore, I recommend acceptance of this paper.

---

### Decision · Program_Chairs · 2025-01-22

Accept (Poster)